# Integrative discovery of treatments for high-risk neuroblastoma

Elin Almstedt [1], Ramy Elgendy [1], Neda Hekmati[1], Emil Rosén [1], Caroline Wärn[1], Thale Kristin Olsen[2], Cecilia Dyberg[2], Milena Doroszko[1], Ida Larsson [1], Anders Sundström[1], Marie Arsenian Henriksson[3], Sven Påhlman[4], Daniel Bexell [4], Michael Vanlandewijck [1,5], Per Kogner [2], Rebecka Jörnsten[6], Cecilia Krona[1] & Sven Nelander[1]*

Despite advances in the molecular exploration of paediatric cancers, approximately 50% of children with high-risk neuroblastoma lack effective treatment. To identify therapeutic options for this group of high-risk patients, we combine predictive data mining with experimental evaluation in patient-derived xenograft cells. Our proposed algorithm, Target-Translator, integrates data from tumour biobanks, pharmacological databases, and cellular networks to predict how targeted interventions affect mRNA signatures associated with high patient risk or disease processes. We find more than 80 targets to be associated with neuroblastoma risk and differentiation signatures. Selected targets are evaluated in cell lines derived from high-risk patients to demonstrate reversal of risk signatures and malignant phenotypes. Using neuroblastoma xenograft models, we establish CNR2 and MAPK8 as promising candidates for the treatment of high-risk neuroblastoma. We expect that our method, available as a public tool (targettranslator.org), will enhance and expedite the discovery of risk-associated targets for paediatric and adult cancers.

[1] Department of Immunology, Genetics and Pathology, Uppsala University, SE-751 85 Uppsala, Sweden. [2] Childhood Cancer Research Unit, Department of Women's and Children's Health, Karolinska Institutet, SE-17176 Stockholm, Sweden. [3] Department of Microbiology, Tumor and Cell Biology, Karolinska Institutet, SE-171 77 Stockholm, Sweden. [4] Division of Translational Cancer Research, Department of Laboratory Medicine, Lund University, SE-223 81 Lund, Sweden. [5] Department of Medicine, Integrated Cardio-Metabolic Centre Single Cell Facility, Karolinska Institutet, SE-17177 Stockholm, Sweden. [6] Mathematical Sciences, Chalmers University of Technology, Gothenburg SE-41296, Sweden. *email: sven.nelander@igp.uu.se

Neuroblastoma is a cancer of the sympathetic nervous system, which accounts for 7% of all childhood cancers and 15% of childhood cancer-related deaths[1]. Mostly diagnosed before 2 years of age, it can lead to a broad range of clinical outcomes, from spontaneous regression of the tumour to metastatic disease with poor prognosis[2,3]. Clinical manifestations and genetic markers help identify the children with high-risk disease and guide the treatment[3]. Sequencing of neuroblastoma genomes has uncovered actionable mutations, particularly *ALK*, present in 8% of neuroblastoma cases[4,5]. Still, the clinical management of high-risk cases remains difficult, with survival rates <50%[6].

The goal of this study is to identify additional therapeutic targets against high-risk neuroblastoma. To achieve this, we combine integrative data analysis with experimental evaluation in cell lines from patient-derived xenografts. Specifically, we explore the hypothesis that the clinical risk of a neuroblastoma patient, as defined by established clinical and genomic markers, can be well approximated by an mRNA signature. By integrating such signatures with recent pharmacogenomic data from cancer and non-malignant cell lines[7] and drug-to-target networks[8], we then propose that such high-risk signatures can be associated to drugs that share common and statistically enriched therapeutic targets. Supporting this idea, recent work on neuroblastoma has identified risk signatures defining differentiation, patient outcome, and 1p36.3 deletion[9–12]. Such signatures may reflect important druggable processes in tumour cells that are not readily implied by standard DNA sequencing (c.f. *ALK*). Integrative modelling across data sets enables the detection of genetic and epigenetic regulators of risk signatures[13–15]. Predicting drug targets for high-risk neuroblastoma with broad-scope integrative algorithms thus seems a promising strategy that has not yet been explored. We expect such large scale integration to be particularly productive when based on RNA data, partly because RNA signatures are well supported as indicators of neuroblastoma disease processes, but also because it is available in higher quantities than, for instance, protein profiling data.

When the mRNA signature is known, search tools[16–19] can be used to propose drugs that match the gene markers. By comparison, performing an analysis that links clinical risk factors and disease signatures to protein targets is an analytical task of a different dimension, and requires additional integration and analysis steps. Therefore, a second goal of this work is to provide a generally applicable tool (not specific to neuroblastoma) that will facilitate the association between cancer risk factors, signatures and therapeutic targets.

We propose an algorithm, TargetTranslator, to systematically identify targets for high-risk neuroblastoma. Using data from US and EU patient cohorts, our algorithm finds 10 mRNA signatures of neuroblastoma risk and differentiation, which are mapped to 19,763 unique compounds in 14 cell line models, revealing 88 statistically significant protein targets against high-risk neuroblastoma. We then characterise selected targets by more than 700 RNA profiling experiments in drug-treated neuroblastoma cells and show that interfering with two drug targets, the mitogen-activated protein kinase 8 (MAPK8) and the cannabinoid receptor 2 (CNR2) suppress tumour growth in both zebrafish and mouse xenograft models. Together, these results deepen our understanding of neuroblastoma vulnerabilities and provide a tool for data-guided cancer target discovery.

## Results

**Discovery approach and data sources.** Previous work provides a strong rationale for exploring neuroblastoma by RNA signatures, which can serve as proxy indicators of oncogene activation, patient risk, and tumour cell differentiation status[9–12]. The overriding goal of our strategy is, therefore, to identify targets with potential to modulate such RNA signatures and thereby suppress malignant phenotypes of neuroblastoma cells, potentially resulting in a reduction of tumour growth. Exploring this idea, our discovery approach has two key steps. First, we estimate mRNA signatures of neuroblastoma from patient omics data, which are optimised for integration with public data sources to predict therapeutic targets (Fig. 1a). Second, we evaluate the predicted targets in patient-derived neuroblastoma cells (Fig. 1b). Specifically, the experiments evaluate (i) if mRNA disease signatures change as predicted after drug treatment, (ii) if treatment affects malignant phenotypes in cells, and (iii) if treatment inhibits in vivo tumour growth. To facilitate the evaluation of multiple targets at a reasonable cost, we propose two methods: 384-well SMART-Seq2-based RNA sequencing (RNA-Seq), and automated imaging of GFP-tagged cells in zebrafish embryos (Fig. 1b).

We integrated three different levels of large scale data (Fig. 1a). The first level of data comprised neuroblastoma omics data from the R2[20], NIH-TARGET[21] and SEQC[22] biobanks, with a total of 833 cases. From the clinical, genetic and transcriptional parts of these data, we established 16 disease-associated risk factors, oncogenes, and signatures (Table 1, Supplementary Data 1). Among these, stage, age, *MYCN* and 11q deletion are routinely used for clinical management[3,23], and *ALK* mutation for targeted therapy[24]. We also added gene signatures of patient risk[11], oncogene activation[25] and differentiation level[9,12]. (Because they were not genotyped in all three data sets, mutations of *TERT* and *ATRX* were not part of the analysis.) The two other levels of data were pharmaco-transcriptomic data from the LINCS/L1000 database of drug-induced mRNA changes in human cells[7] and drug-to-protein target information from the STITCH5 database[8]. To gain predictive power, we used a version of the LINCS/L1000 data, in which the transcriptional effect of a drug is estimated from multiple replicates (Supplementary Fig. 1). The full data set thus comprised data for 833 cases, annotated with 16 risk factors, oncogenes and disease signatures, mRNA drug response data for 19,763 unique chemical compounds (we will use the term 'drug' below, for a more concise presentation) and 452,782 links between drugs and protein targets, involving 3421 unique LINCS/L1000 drugs and 17,086 unique targets.

**Association between risk factors, signatures and targets.** Our algorithm, TargetTranslator, estimates mRNA signatures by solving a linear least squares problem, in which each risk factor (e.g. *MYCN* amplification) or genetic aberration is fitted by linear weights (i.e. the signature) to match the expression levels of the 978 genes in the LINCS/L1000 data (Eqs. (1)–(3) in Methods, and Supplementary Figs. 1 and 2). Applying this method to the neuroblastoma data, we confirmed the quality of the fitted signatures by cross-validation, whereby we checked the consistency (correlation) of signatures between the three different cohorts. For example, signatures of *MYCN* amplification estimated from each of the R2, TARGET and SEQC cohorts were all highly correlated, with an average Pearson correlation ($r$) of 0.86. Out of our 16 mRNA signatures, 10 were correlated (average $r > 0.4$; Fig. 2a), and were thus considered robust and used for further analyses. The consistency across cohorts was higher with our method than using the previously reported Characteristic Direction[26] algorithm (Fig. 2a). We conclude that many, but not all, clinical, genetic or transcriptional markers associated with neuroblastoma are well approximated by TargetTranslator signatures fitted to the L1000 genes. A principal component analysis (PCA) of our RNA signatures also indicate that they separate largely into differentiation (PC2) and risk-related (PC1) factors (Fig. 2b).

Mathematically, target discovery is done in two steps. TargetTranslator first computes a match score (between 0 and 1)

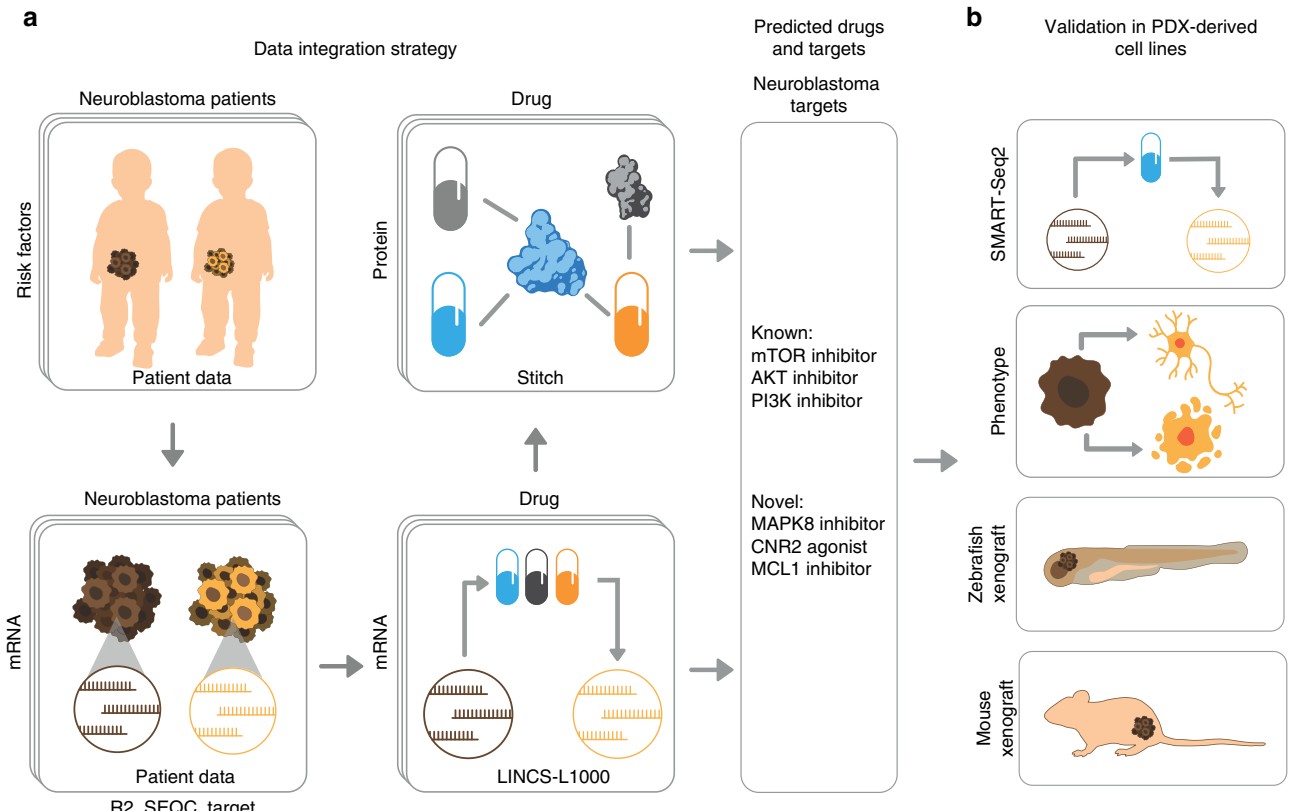

**Fig. 1 Integrative discovery of treatments for high-risk neuroblastoma. a** Step 1: Data integration. We combined omics data from three cohorts of high-risk neuroblastoma (R2, TARGET and SEQC) to construct risk signatures, which were linked to pharmaco-transcriptomic (L1000) and drug target (STITCH) data, resulting in associations between disease signatures and therapeutic targets. **b** Step 2: Experimental evaluation. Using tumour cells from high-risk cases, we combine RNA sequencing, cell-based assays, and animal models to confirm targets.

**Table 1 Clinical data and signatures used for target predictions.**

| Clinical/genetic | Description | Gene signatures | Description | Signature source |
|---|---|---|---|---|
| COG | Risk classification system | Differentiation 1 | Adrenergic vs mesenchymal | van Groningen et al.[9] |
| INSS | Neuroblastoma stage | Differentiation 2 | Nuclear hormone receptors | Ribeiro et al.[12] |
| Age | Age of patient | Differentiation 3 | Selected differentiation markers | See Supplementary Data 1 |
| Risk score | Cox regression risk score | Risk | Signature of survival outcome | De Preter et al.[11] |
| *MYCN* amp | *MYCN* amplification | 1p36 RNA | Signature of 1p36 deletion | White et al.[10] |
| *ALK* mut | *ALK* mutation | *ALK* | Signature of *ALK* mutation | Lambertz et al.[25] |
| 11q del | 11q deletion | 11q RNA | Genes on chromosome 11q | Molecular Signatures Database |
| 17q gain | 17q gain | 17q RNA | Genes on chromosome 17q | Molecular Signatures Database |

between each neuroblastoma signature and the drug-induced expression profiles in LINCS/L1000 (Eq. (4) in Methods). The scores (one score per drug) are summarized to an empirical cumulative distribution function (ecdf) plot and their significances are assessed via permutation tests (Fig. 2c). For each protein target in STITCH5, we then separate the drugs in LINCS/L1000 into two groups: those that have a link to the target and those that do not (by link, we mean a STITCH score above a stringent threshold, see Methods). A two-sample, one-tailed, Kolmogorov–Smirnov test is subsequently used to compare scores for the two groups of target-associated vs non-associated compounds (Fig. 2d). Proteins for which the null hypothesis is rejected are predicted to be targets. We correct for multiple testing across targets and report FDR adjusted $q$-values. This procedure is repeated twice, with alternating sign ($-$/$+$) of the signature, to identify targets likely to be associated with suppression ($-$) and enhancement ($+$) of each signature. Both cases can be therapeutically relevant since we both seek to

suppress signatures associated with risk (like *MYCN*) and enhance signatures likely to be associated with slower tumour growth (like differentiation).

**Identification of 88 targets for high-risk neuroblastoma.** Following this strategy, TargetTranslator gave a rich set of predictions of druggable targets for high-risk neuroblastoma. Applied to the three neuroblastoma cohorts, we obtained a total of 88 enriched drug targets that have a $q$-value of <0.0001 for at least one risk factor (Fig. 3). First, we noted that already-established targets, including MTOR and PIK3 isoforms[27], EGFR[28] and CDK2[29], were associated with several of the molecular and clinical risk signatures, as indicated by a shift in the ecdf curve (Fig. 3, example of MTOR in Fig. 2c). Second, there was a spectrum of targets that have limited evidence of neuroblastoma effect in the literature which still had a strong enrichment. Notable examples include MAPK8 and the cannabinoid receptor

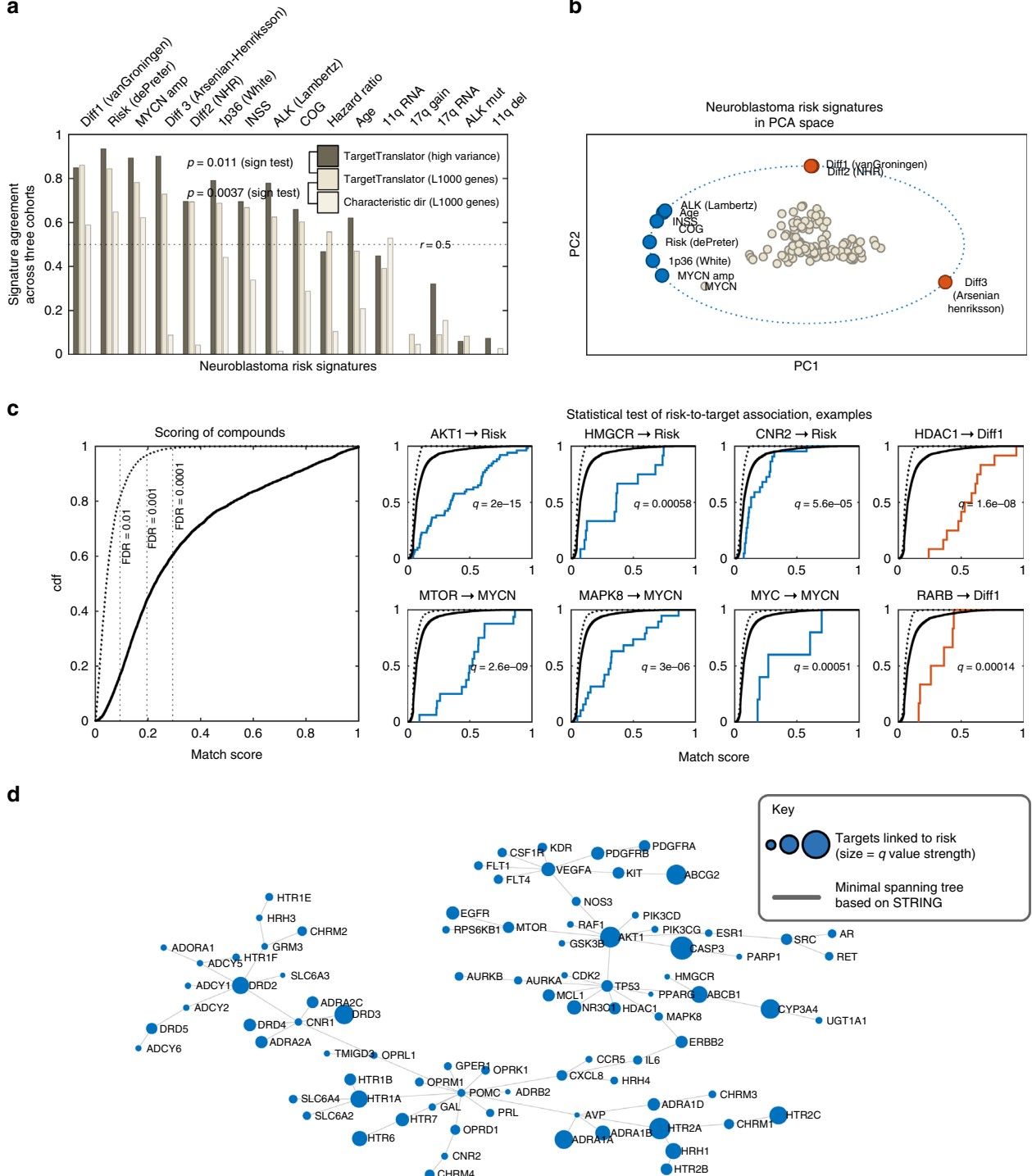

**Fig. 2 Detection of multiple targets linked to neuroblastoma risk factors. a** Validation of neuroblastoma risk signatures by each signature's agreement across three independent cohorts (R2, TARGET and SEQC). Dark grey = signatures constructed by TargetTranslator, using high variance genes; grey = TargetTranslator, L1000 landmark genes; white = L1000 landmark genes, using the Characteristic direction algorithm. **b** Principal component analysis of the 10 most reproducible signatures, showing risk factors at unit length and marker genes as points; note collinearity between signatures and the distinct/ opposing direction of differentiation signatures. Blue: signatures associated with disease pathways or poor outcome. Red: signatures associated with differentiation. **c** Left: Matching compounds and drugs in L1000 to neuroblastoma signatures; score empirical cumulative distribution function (ecdf) for 19763 drugs, based on L1000 from 14 cell lines (dashed line = permutation control). Right: Detection of enriched targets by shifts in the ecdf curve. L1000 compounds were mapped to drug targets using the STITCH database. Examples of drugs with a common target are highlighted (coloured curve). Note examples of confirmed (*AKT1, MTOR, HDAC1*), predicted (*HMGCR, MAPK8, CNR2*) and control targets (*MYC, RARB*, associated to *MYCN* and differentiation signatures, respectively). *q* are FDR-controlled *p*-values of a Kolmogorov–Smirnoff test that compares target-specific compounds (coloured curve) vs other compounds. **d** Visualization of pathway dependencies for all targets with *q*-value < 0.0001, mapped to the minimal spanning tree (MST) of all STRING links between targets. (MST facilitates visualization, by removing redundant links).

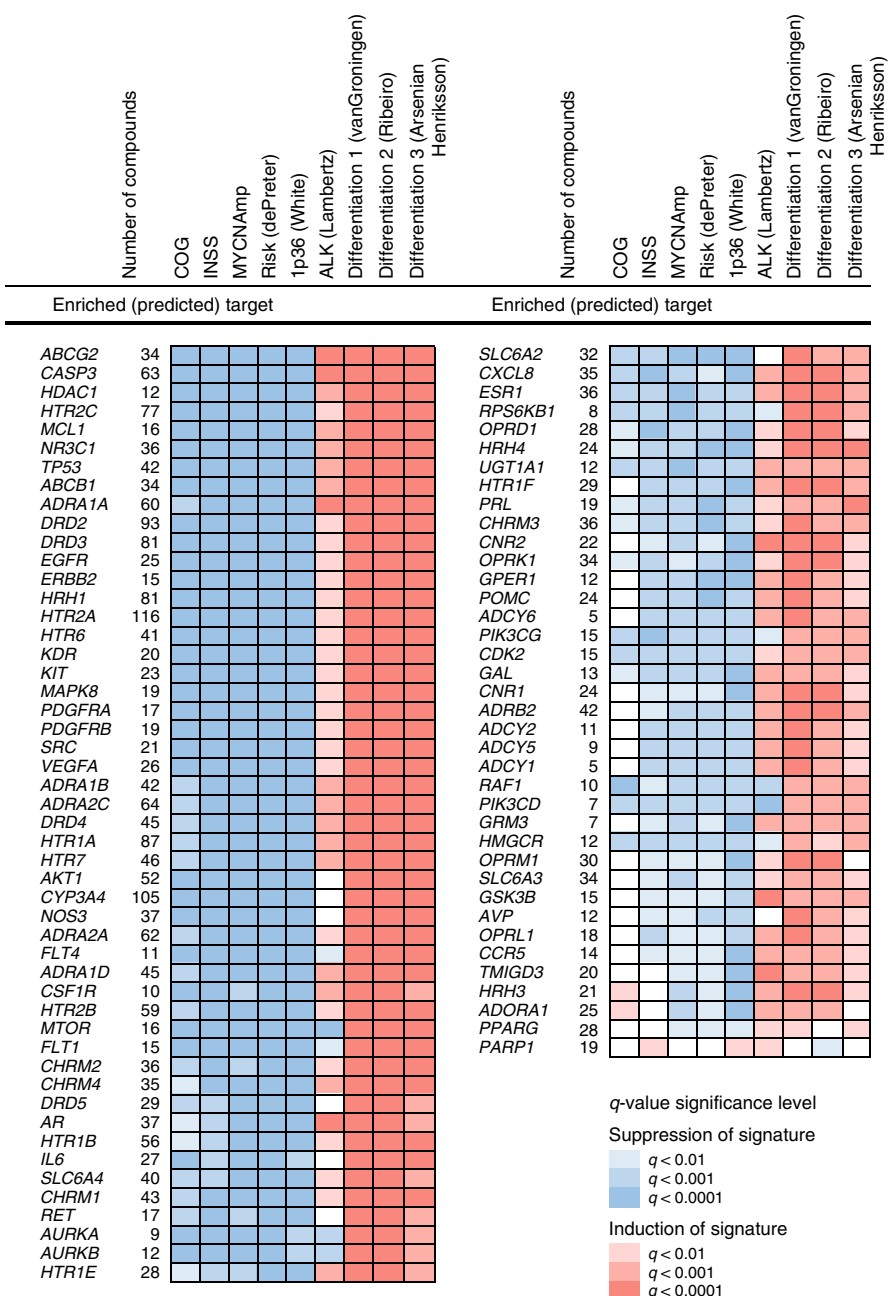

**Fig. 3 Drug targets predicted by TargetTranslator for neuroblastoma signatures.** 88 drug targets predicted by TargetTranslator. Red: target is associated with induction of signature; Blue: target is associated with suppression of signature. Shades represent strength of q-value.

CNR2 (Fig. 2c), as well as MCL1 (Fig. 3). As further validation of our method, we found that MYC was detected as a target selectively associated with the *MYCN* amplification signature and that the RARB receptor of retinoic acid (which induces a differentiation phenotype in neuroblastoma[30]), was significantly associated to differentiation signatures (Fig. 2c). Inspecting the results further, we also found a number of interesting drugs, which had a high ranking match score for at least one risk factor, but where LINCS/L1000 contained too few similar drugs (fewer than 4 with the same STITCH5 target) to motivate target enrichment with the Kolmogorov–Smirnov test. Notable examples were drugs targeting glycosylceramide synthase UGCG (DL-PDMP), the benzodiazepine receptor TSPO (PK11195) and ROCK (fasudil).

Taken together, TargetTranslator detected targets and small molecules that merit further consideration for neuroblastoma therapy. The detected targets show differential association with molecular risk and differentiation signatures (Fig. 3). Minimal spanning tree (MST) analysis of highly scoring drug targets visualized possible relationships between predicted targets (Fig. 2d), suggesting possibilities for single-agent or multiple agent interventions.

**RNA profiling confirmed TargetTranslator predictions.** To explore the therapeutic potential of our predictions, we selected a set of representative inhibitors or agonists against 11 targets for experimental evaluation (Table 2a). Of these, four were known targets and under clinical development: PI3K (trial NCT03458728), MTOR (NCT03213678), CDK4/6 (NCT01747876, NCT02644460) and HMGCR (NCT02390843). Three, in turn, show some promise in preclinical animal studies (PPARG[31], CDK2[29], ROCK[32] and four were comparatively less studied (MAPK8, CNR2, UGCG, TSPO).

**Table 2 Compounds used to evaluate TargetTranslator predictions.**

| Compound | Ligand of enriched target | Top 50 hit | Reference compound | Target |
|---|---|---|---|---|
| a |  |  |  |  |
| AS601245 | x | x |  | MAPK8 and MAPK9,10 |
| Torin-2 | x | x |  | MTOR |
| GW405833 | x | x |  | CNR2 agonist |
| Omipalisib | x | x |  | PIK3 |
| AZD5438 | x | x |  | CDK2 |
| Lovastatin | x | x |  | HMGCR |
| Rosiglitazone | x | x |  | PPARG |
| Fasudil |  | x |  | ROCK |
| DL-DPMP |  | x |  | UGCG |
| PK11195 |  | x |  | TSPO |
| JQ1 | x |  | x | MYCN, MYC |
| Retinoic acid | x |  | x | RARA, RARB |
| b |  |  |  |  |
| JWH133 | x |  | x | CNR2 agonist |
| HU308 | x |  | x | CNR2 agonist |
| ACEA | x |  | x | CNR1 agonist |
| SR144528 | x |  | x | CNR2 antagonist |
| JNK-IN-8 | x |  | x | JNK inhibitor (MAPK8,10 selective) |
| SP600125 | x |  | x | JNK inhibitor (MAPK8,9,10) |
| CC-930 | x |  | x | JNK inhibitor (MAPK9,10 selective) |

Compounds used to evaluate predictions (a) and to assess CNR2 and MAPK8 as targets (b). 'x' indicates that a compound is a ligand of an enriched target, a top 50 hit for at least one risk signature, or included as a reference compound

In a first experiment, we tested if neuroblastoma cells treated with these compounds would change their gene expression pattern as predicted (i.e. by suppression or induction of the risk signatures, c.f. Fig. 3). As a representative model, we selected two patient-derived xenograft (PDX) cultures from two high-risk patients with *MYCN* amplified neuroblastoma, termed NB-PDX2 and NB-PDX3. Both cell lines were treated with 13 drugs (the 11 targeted drugs above, plus the differentiation agent retinoic acid and the BET bromodomain inhibitor JQ1, which downregulates *MYCN*[33]), at three concentrations equivalent to IC10, IC20 and IC50, for 6 and 24 h, followed by RNA profiling using smart-Seq2 (Fig. 4a). The drug-induced transcriptional changes in NB-PDX2 and NB-PDX3 cells were observable as distinct directions in the principal component space (Fig. 4b). The magnitude of transcriptional change along the principal components was proportional to drug dose and drug exposure time (Supplementary Table 1); drug-induced vectors were also relatively conserved between the NB-PDX2 and NB-PDX3 cultures (Supplementary Fig. 3). In both cell cultures, MTOR/PI3K and CDK4/6 inhibitors (omipalisib, torin-2 and palbociclib) induced a change along PC1, shifting the transcriptome away from the risk signatures. Lipid metabolism targeted agents (lovastatin, DL-PDMP and rosiglitazone) had high magnitudes along PC2, and PC3 was led by MAPK8 inhibitor AS601245. PC 4, finally, contained the differentiation-inducing agents RA and JQ1, shifting the transcriptome towards the differentiation signatures.

To formally assess the accuracy of our predictions, we performed a receiver operating characteristic (ROC) analysis. First, we identified all ($n = 24$) cases in which one of the 11 drugs either induced ($n = 8$) or suppressed ($n = 16$) a risk signature in the neuroblastoma patient-derived cells. Next, we computed the sensitivity and specificity of TargetTranslator with respect to detecting these associations. ROC analysis showed an excellent sensitivity vs specificity curve for both negative associations (vs no association) and positive associations (vs no association), with an average area under the curve (AUC) of 0.916 (Fig. 4c). The reproducibility of the scores was on a comparable level when contrasting the two PDX cultures (AUC 0.848, Supplementary Fig. 3E, F).

Pathways affected by the tested compounds were consistent with drug mechanism of action. Specifically, Gene Set Enrichment Analysis (GSEA) revealed that cholesterol homoeostasis was altered upon lovastatin and rosiglitazone treatment, the suppression of MYC targets by JQ1 and the inhibition of MTOR pathway members following torin-2 treatment (Fig. 4d). Most of our predictions, including the CNR2 inhibitor GW405833, the UGCG inhibitor DL-PDMP, the TSPO antagonist PK11195, and the MAPK8/JNK inhibitor AS601245, downregulated the expression of genes induced during the cell cycle, implicating effects on neuroblastoma cell growth (Fig. 4d).

Together, these data show that the effects in neuroblastoma cells were dose-dependent, correlated well with our predictions, consistent with known targets and affected therapeutically relevant pathways.

**Predicted compounds suppress malignant phenotypes**. The observed transcriptional changes suggested that treatment with our predicted compounds should also produce phenotypic effects. Indeed, all 11 tested compounds suppressed viability of neuroblastoma cultures, as determined by dose–response curves after 72 h of drug exposure. For 10 compounds, the reduction in viability was more marked in neuroblastoma cells (NB-PDX2, NB-PDX3, SK-N-BE(2) and SK-N-SH) when compared with glioblastoma reference cells (U3013MG), the difference being largest for the CNR2 modulator GW405833 (Fig. 5a). Using the BET inhibitor JQ1, which suppresses *MYCN* transcription[33], and the differentiation agent retinoic acid as positive controls, we found that reduced viability coincided with an induction of apoptosis markers for seven compounds, as observed by live-cell monitoring (Fig. 5b, c).

The predicted compounds also produced significant effects on N-Myc protein levels and cell differentiation, both of which are prognostic in neuroblastoma patients[3] and are mechanistically linked[34–36]. Specifically, N-Myc protein was suppressed by all compounds except torin-2, omipalisib and rosiglitazone (Fig. 5d, e). Neurite outgrowth—a proxy for differentiation—was observed following treatment by four compounds (palbociclib, DL-PDMP,

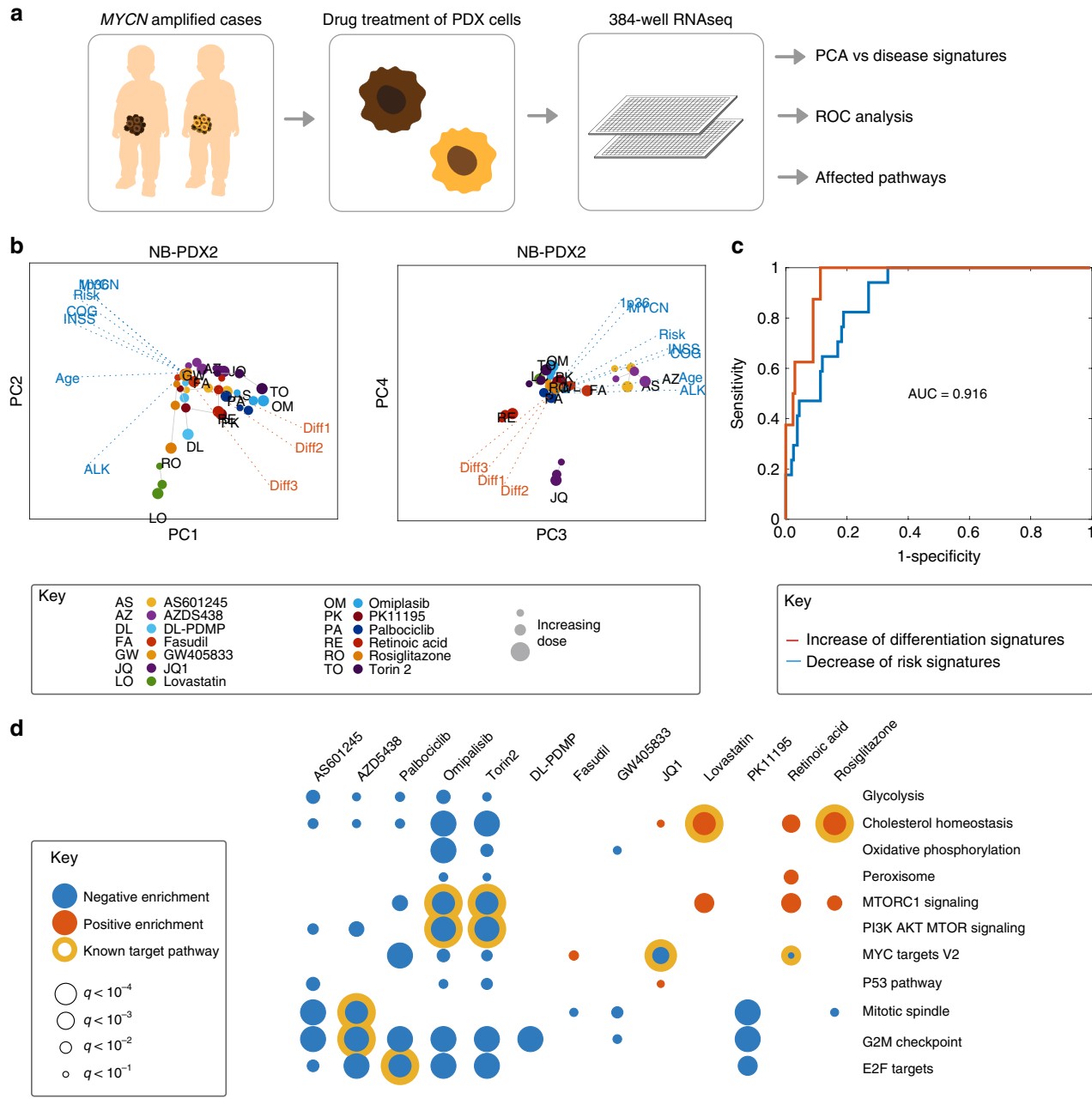

**Fig. 4 RNA profiling confirmed modulation of neuroblastoma risk signatures. a** Experiment concept. **b** Principal component analysis of results for patient 1 (NB-PDX2). Note similar directions of PI3K and MTOR inhibitors omipalisib (OM) and torin-2 (TO) along PC1, and differentiation-inducing agents retinoic acid and JQ1 along PC 4. See also Supplementary Fig. 3. **c** ROC curve analysing the predictive power of TargetTranslator, defined as the ability to predict (from the public data sources) whether a particular gene signature will decrease (blue) or increase (red) after drug treatment; showing average for 10 risk signatures and all tested drugs, overall area under the curve correctness of 0.916. **d** Gene Set Enrichment Analysis (GSEA) of drug-induced changes, showing suppressed (blue) and induced (red) pathways. Yellow circles mark the known action of each corresponding drug.

lovastatin and GW05833) as well as the positive control retinoic acid (Fig. 5f, g).

Jointly, these results support the conclusion that compounds against predicted targets were active against in vitro growth of neuroblastoma cell lines and patient-derived cultures, with (for some compounds) concurrent effects on N-Myc protein levels and differentiation.

**Cellular action of CNR2 agonists and MAPK8 antagonists.** The finding that a CNR2 agonist GW405833 and a MAPK8 inhibitor AS601245 were active in neuroblastoma cells, motivated

experiments to evaluate their on-target selectivity. For cannabinoid receptors, a number of agents are available to target both CNR2[37] and its pharmacologically relevant paralog CNR1[38,39] (Table 2b). In experiments on NB-PDX3 cells, two additional specific CNR2 agonists, JWH133 and HU308, reduced neuroblastoma cell viability, whereas the CNR1 agonist ACEA did not (Supplementary Fig. 4A), nor did antagonists of either CNR1 or CNR2 suppress neuroblastoma cell growth (Supplementary Fig. 4B). We further found that neuroblastoma cells that were pre-treated with a CNR2 antagonist (SR144528) were protected against GW405833 ($p = 0.014$, Student's $t$-test), while pre-treatment with a CNR1 antagonist (otenabant) had no

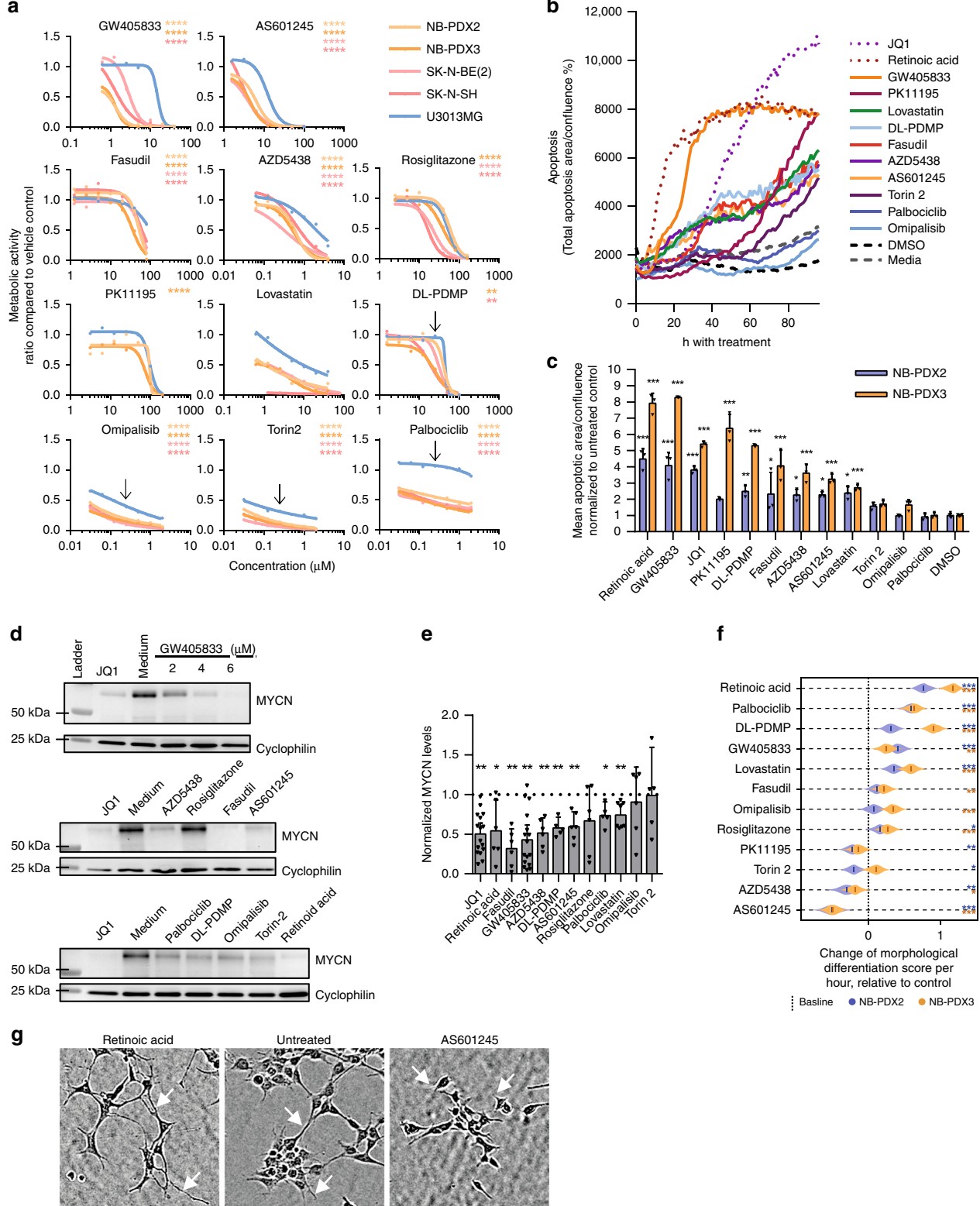

protective effect (Supplementary Fig. 4C, D). Together, these findings were consistent with CNR2 activation as the key molecular pharmacological mechanism.

We similarly investigated the effect of multiple agents targeting MAPK8 and its pharmacologically relevant homologues MAPK9 and MAPK10, against which AS601245 has reported activity. Neuroblastoma cells were sensitive to JNK-IN-8[40] and SP600125[41], but not the MAPK9/10 selective compound CC-

930[42], which lacked effect even at high doses (Supplementary Fig. 4E, F). The three compounds with anti-neuroblastoma effect (AS601245, JNK-IN-8 and SP600125) have higher affinity for MAPK8 than at least one of the other isoforms, while CC-930 has a 12-fold lower affinity for MAPK8 than the other isoforms (Supplementary Fig. 4F). As a complementary analysis based on genetic methods, we considered a public data set of RNA interference data from the Broad institute DepMap[43], in which

**Fig. 5 Predicted targets suppressed malignant phenotypes in patient-derived neuroblastoma cells. a** Viability response of four neuroblastoma (red) and one glioblastoma (blue, U3013MG) cell lines after 72 h of treatment. Asterisks indicate the level of significance for each neuroblastoma cell line compared with U3013MG. (When applicable, IC50 was used for statistical comparisons, otherwise, the dose is indicated by the arrow.) **b, c** Apoptotic response (cleaved CASP3/7) of each compound (mean, $n = 3$) and comparison of compounds at 96 h time point (mean, standard deviation). **d** Reduction of N-Myc levels after 48 h drug exposure at IC50 concentrations, cropped image. The full image is found in the Source Data file. **e** Quantified N-Myc levels for both NB-PDX2 and NB-PDX3 (mean, 95% confidence interval; JQ1, GW, $n = 16$; lovastatin, $n = 8$; omipalisib, RA, AZD5438, rosiglitazone, fasudil, AS601245, $n = 6$; palbociclib, DL-PDMP, Torin-2, $n = 5$; one-sample $t$-test with Benjamini–Hochberg FDR correction). **f** IC10 drug effects of neurite outgrowth for NB-PDX2 (blue) and NB-PDX3 (yellow), bootstrapping estimates ($n = 1000$). A higher morphological differentiation score indicates longer cell protrusions. Stars show significance levels compared with negative control for the respective cell lines. **g** Representative image of cell protrusions (white arrow) after 72 h of treatment. *$p < 0.05$, **$p < 0.01$, ***$p < 0.001$, ****$p < 0.0001$.

the tested neuroblastoma cell lines ($n = 9$) are more vulnerable to MAPK8 knockdown than non-neuroblastoma cell lines ($n = 704$), but not MAPK9 or MAPK10 (Supplementary Fig. 4G). Jointly, these results indicated that MAPK8 is an important protein in the response to AS601245.

**Targeting MAPK8 and CNR2 reduces tumour growth in zebrafish.** Following the promising in vitro results, we evaluated the in vivo potential of the compounds predicted by Target-Translator. First, we implemented a high-throughput zebrafish xenograft assay of neuroblastoma. As the interrenal gland (the fish homologue of the human adrenal gland) is too small for high-throughput and consistent tumour injections, we chose the developing brain as the site to inject PDX-derived neuroblastoma cells. This injection site was also motivated by the fact that 8% of all neuroblastomas and 20% of progressive neuroblastomas spread to the brain[44,45]. NB-PDX3 cells were GFP-tagged using a lentiviral construct, FACS sorted to enrich for highly fluorescent cells and ~150 cells per fish were injected into the midbrain of 1-day post fertilization (dpf) Casper zebrafish embryos (Fig. 6a). This resulted in consistent engraftments with 90% of the tumours localized to the midbrain/hindbrain (Fig. 6b). Neuroblastoma proliferation was not affected by either decrease in temperature or GFP-tagging and xenografts contained MKI67-positive human cells at 5 dpf (Supplementary Fig. 5A–D). GFP signal highly correlated with confluence over time ($n = 15$, Supplementary Fig. 6E) and was used as a proxy for tumour growth.

Next, we determined the tolerance to each compound by exposing 2 dpf embryos to increasing (IC20, IC50 and IC80) doses of the predicted drugs (Fig. 6c). Of the 12 tested compounds, 8 were well tolerated at neuroblastoma-inhibiting doses (AZD5438, omipalisib, palbociclib, fasudil, GW405833, torin-2, AS601245 and positive control doxorubicin) whereas 4 showed signs of toxicity (lovastatin, DL-PDMP, PK11195 and rosiglitazone). The latter group corresponds to drugs implicated as modifiers of lipid metabolism (c.f. PC2 in Fig. 4b), and their toxicity likely reflects essentiality of these processes in this particular model system, as both statins and rosiglitazone are clinically approved drugs and PK11195 has been used as a positron emission tomography (PET) radiotracer in humans[46].

Focusing on the CNR2 and MAPK8 targeted compounds, zebrafish xenografts were treated for 72 h by adding drug (GW405833 or AS601245) to the surrounding water. Before (2 dpf) and after (5 dpf) treatment, zebrafish were imaged both dorsally and laterally using a Vertebrate Automated Screening Technology (VAST) system (Fig. 6d), and each fish was scored for the increase in GFP signal. MAPK8 inhibitor AS601245 inhibited tumour size to 21% of DMSO treated control after 3 days ($p < 0.0001$, Dunnett's multiple comparison test), while GW405833, which was the most neuroblastoma-selective inhibitor in vitro, reduced the tumour size to 75% of control ($p = 0.0335$, Dunnett's multiple comparison test (Fig. 6e, f). The lower temperature in the assay did not potentiate NB cells to either

compound (Supplementary Fig. 5F). Together, these first in vivo results supported TargetTranslator as a viable tool for predicting neuroblastoma targets with significant observed effects on tumour size in a zebrafish model.

**CNR2 agonist GW405833 reduces tumour growth in mice.** To evaluate the in vivo effect of AS601245 and GW405833 in a mammalian model, we performed a treatment study on the *MYCN* amplified SK-N-BE(2) flank-injected mouse xenografts. Mice were monitored during 8 days of treatment, with assessment of tumour growth rate, tumour weight after 8 days and immunohistochemistry as endpoints (Fig. 7). After 8 days, GW405833 significantly reduced tumour size to 75% of vehicle-treated animals ($p = 0.034$, linear mixed model), with no indications of toxicity in any of the 10 treated mice (Fig. 7a–e). Consistent with the in vitro results (c.f. Fig. 5b–c), GW405833 caused elevated apoptosis in treated tumours, measured by immunohistochemistry of cleaved PARP (Fig. 7f–h). AS601245, and reduced growth rate ($p = 0.048$, linear mixed model), but showed signs of toxicity in 3 of 8 mice, after which the treatment group was discontinued and remaining mice were sacrificed (Supplementary Fig. 6). Together, these data provided an independent set of data to support the potential of a CNR2 agonist against neuroblastoma, and encourage investigation to define MAPK8 (JNK1) inhibitors with improved toxicity profile as a future direction.

**The TargetTranslator tool.** So far, we have used our method to study neuroblastoma; to facilitate the application of Target-Translator to other cancer diagnoses as well, we have implemented a web version of TargetTranslator, which comprises a simple workflow, in which users (i) select data sets of interest, (ii) identifies risk factors and signatures of interest and (iii) run the pipeline to find risk-associated drugs and therapeutic targets. The current implementation enables users to upload their own data but also lets users analyse pre-uploaded data sets of R2, TARGET, and SEQC neuroblastoma patient cohorts and an additional 33 clinically annotated cancers from the cancer genome atlas (TCGA) consortium[47]. TargetTranslator is also available as a standalone R package, intended for expert users.

## Discussion

This study aimed to identify treatment options with in vivo relevance for high-risk neuroblastoma. A tailored algorithm, Target-Translator, estimated robust mRNA signatures of neuroblastoma, which were associated with drug targets. These, in turn, were characterized by multiplexed RNA-Seq, tested in cells and xenograft models, and resulted in leads for neuroblastoma treatment. Possible applications include target discovery for tumours other than neuroblastoma, particularly to discover druggable cancer targets not found by standard methods such as DNA sequencing.

Taken together, our computational and experimental results strongly support neuroblastoma therapy development through

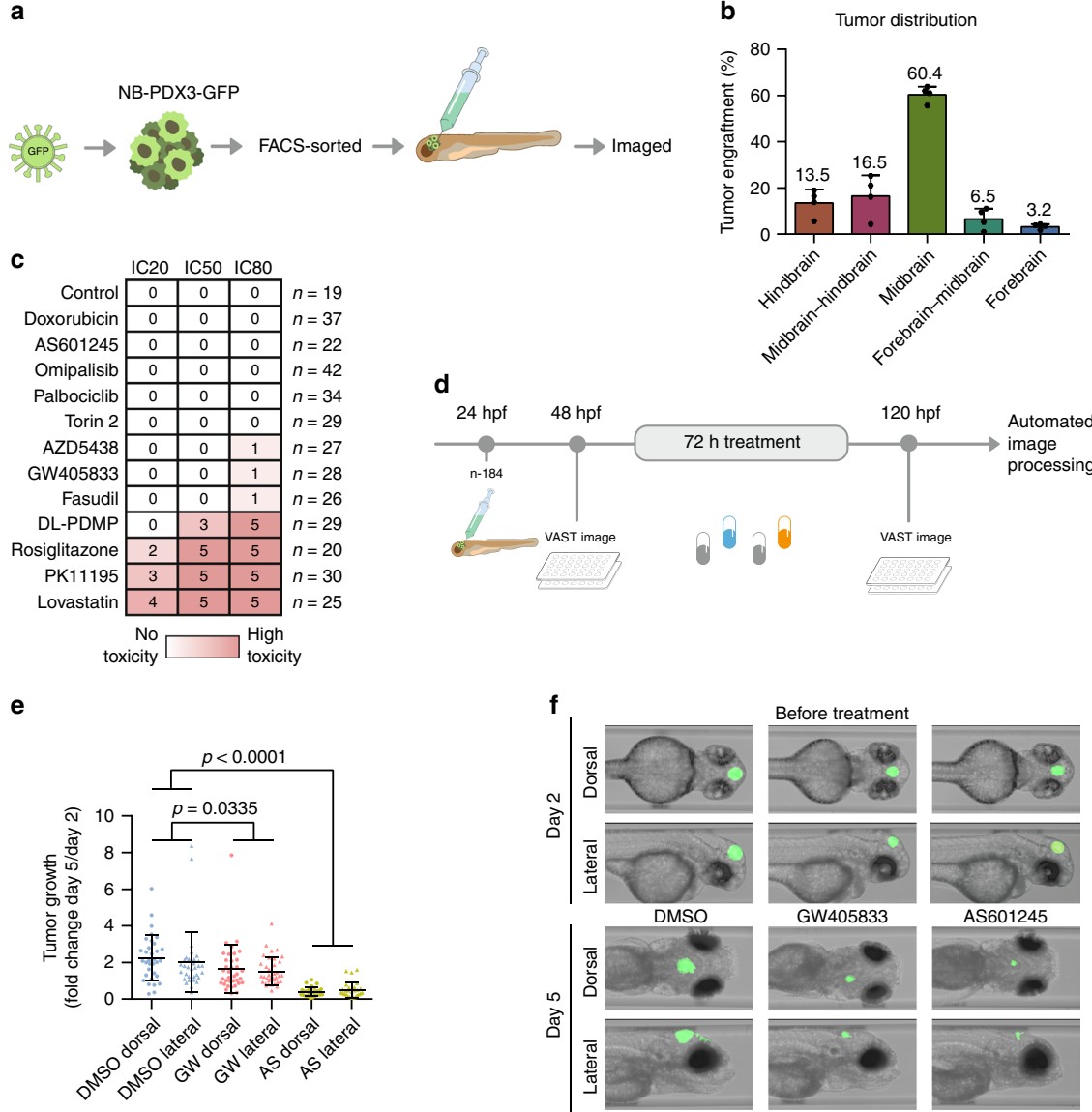

**Fig. 6 Targeting MAPK8 and CNR2 suppressed neuroblastoma xenografts in zebrafish. a** Workflow: patient-derived neuroblastoma cells were tagged with green fluorescent protein (GFP), sorted, and injected into the midbrain of 1-day post fertilization (dpf) zebrafish embryos. **b** Tumour localization 24 h following injection (n = 266) (mean, standard deviation). **c** 2 dpf zebrafish embryos were exposed to drug concentrations corresponding to IC20, IC50 or IC80. Toxicity was noted after 24 h, and score between 0 (no toxicity, white) and 5 (instant, lethal toxicity, red). **d** Automated image-based assay of tumour growth in xenotransplanted zebrafish embryos. **e** Tumour area increase from 2 to 5 dpf (mean, standard deviation). **f** Representative image of the same zebrafish embryos before and after treatment.

RNA risk signature modulation (Fig. 4). The observed transcriptional changes were concurrent with the suppression of malignant phenotypes associated with elevated risk in patients (N-Myc protein levels, proliferation), and induction of favourable features (cellular differentiation, the onset of apoptosis). Conceptually, we remark that using mRNA signatures to approximate risk factors rests on specific assumptions that must be met. Above all, we assume a causative link between gene expression in tumours and observable disease outcome. One example where this is true is *MYCN* amplification, which is causatively linked to neuroblastoma outcome, in a manner that depends on mRNA expression[48]. The Differentiation 1 signature of adrenergic (ADRN) cells, here linked to several targets, also has a well understood mechanistic basis[9]. A second key assumption is that modulation of mRNA levels is possible, either by suppressing a signature driven by an oncogene, such as *MYCN*, or shifting a

plastic cell state towards a more favourable one, such as forcing differentiation. When these conditions are met, TargetTranslator makes logical sense as a tool for target discovery. By contrast, in the minority of cases where there is no reproducible RNA signature (i.e. limited consistency between cohorts, Fig. 2a), further exploration will be needed. For instance, while localized oncogene amplification of *MYCN* yields consistent signatures, our analysis speaks against the use of broad chromosomal regions, like 11q and 17q. TargetTranslator also depends on a sufficient number of patient cases to produce robust signatures, as seen for *ALK* mutation status in this study. In these particular cases, better definition of functionally important markers, and more extensive patient cohorts would be needed to gain analytical power.

Our in vivo results highlight a possible therapeutic potential of two targets: MAPK8/JNK1 and the cannabinoid receptor 2 (CNR2). Signalling pathways upstream of MAPK (receptor tyrosine kinases

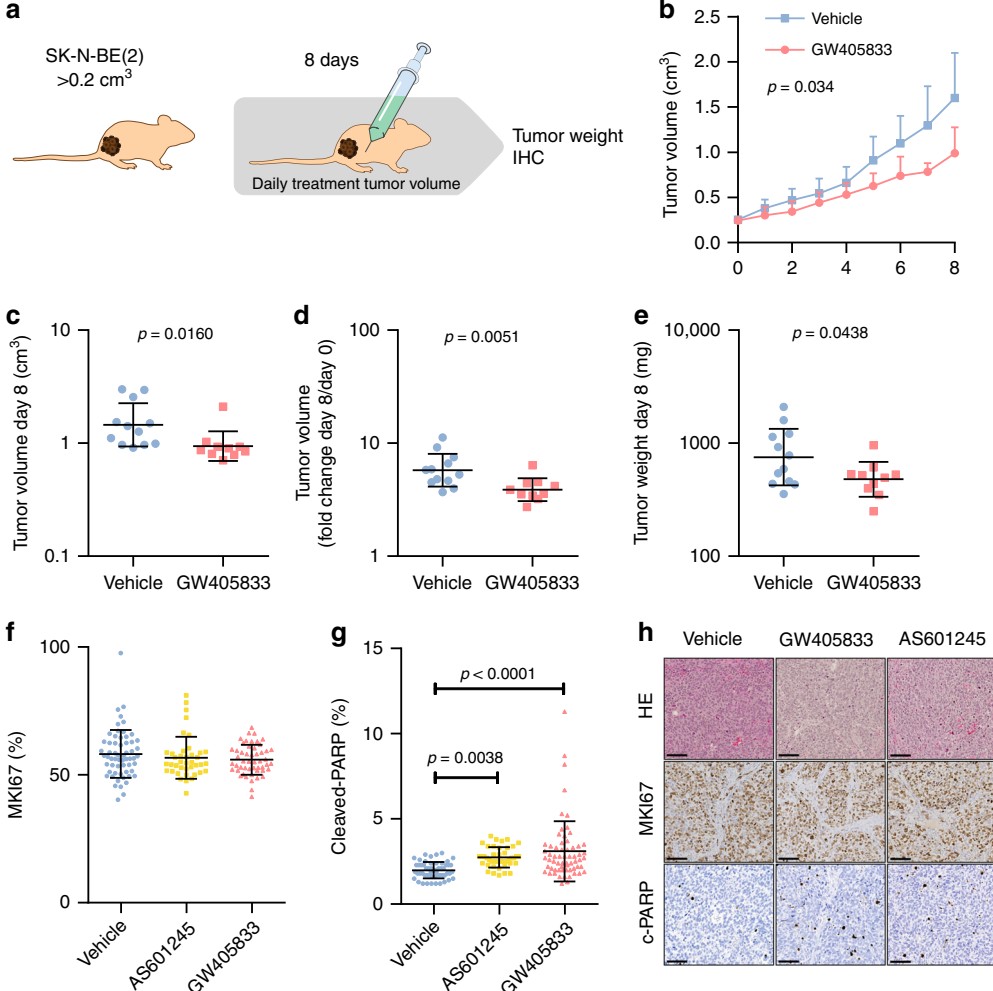

**Fig. 7 GW405833 reduces neuroblastoma growth in vivo. a** Mice were engrafted with $15 \times 10^6$ SK-N-BE(2) cells s.c. and randomized to receive a daily i.p. injection of GW (45 mg/kg; $n = 10$) or vehicle ($n = 12$) for 8 days, starting at the appearance of palpable tumours of >0.2 cm³. **b** GW405833 significantly impaired the growth of established human tumours (hierarchical linear model). **c** Point comparison of day 8 tumour volume. **d** Tumour volume increase from day 0 to day 8. **e** Post-mortem tumour weight after 8 days of treatment. **f** Cell proliferation marker MKI67, counted using ImmunoRatio plugin for ImageJ from 10 to 15 representative fields per specimen (DMSO, GW405833, $n = 5$; AS601245, $n = 4$). **g** Apoptosis marker cleaved PARP, counted in using ImmunoRatio plugin for ImageJ from 10 to 15 representative fields per specimen (DMSO, GW405833, $n = 5$; AS601245, $n = 3$). **h** Representative images of tumour histology (HE), MKI67 and c-PARP localization, bar = 100 μm. Statistics: **b** Mean, 95% confidence interval, *p*-value computed from a mixed effects model and corrected for multiple testing using bonferroni correction. **c–g** Mean, standard deviation, Student's *t*-test.

and RAS) are commonly altered in relapse neuroblastoma[49] and activated pathways are correlated with poor prognosis in primary tumours[50]. The role of MAPK8/JNK1 is less well understood, although it has been linked to neuroblastoma progression[51]. Based on our in vivo results, we propose that MAPK8 partly regulates neuroblastoma growth. The cannabinoid receptor CNR2 (encoded by a gene in the commonly deleted 1p36 region) is a relatively unexplored target in cancer. Our computational analysis identified CNR2-targeting compounds linked to multiple neuroblastoma risk signatures (Fig. 2c, Supplementary Fig. 7). Extending results in serum-cultured cells[52], we found that activation of CNR2 by three different agonists reduces viability in patient-derived cell cultures in vitro (Fig. 5a, Supplementary Fig. 4A) and induces apoptosis in vivo (Fig. 7g). Interestingly, we noted that the CNR2 ligand GW405833 activated genes regulated by the rhodopsin-like G-protein coupled receptors (GPCRs), which is a family of GPCRs that includes the cannabinoid receptors (Supplementary Fig. 4H), and the effect on neuroblastoma cells could be blocked by CNR2 antagonist (Supplementary Fig. 4C, D). We speculate that GW405833 acts by downregulation of N-Myc protein leading to

apoptotic onset. A more exact mechanism of action of cannabinoid receptor 2 in high-risk neuroblastoma will be reserved for future work. However, judging from the absence of specific mutations or clear molecular patterns connected to the CNR2 biology in neuroblastoma, atypical targets like CNR2 would most likely be missed by common data mining strategies. This supports the idea that integrated data analysis spanning both multi-patient cohorts and robust statistical methods that detect latent factors in the data will play a crucial role in the future exploration of patient molecular data.

The rapidly growing use of RNA data as a readout in molecular pharmacological experiments motivates further analysis to define how drug action is manifested transcriptionally. Our analysis of selected drugs in neuroblastoma cells revealed consistent drug-specific patterns between cell lines, the magnitude of which was dose-dependent and time-dependent (Fig. 4, Supplementary Table 1). As in previous pharmaco-transcriptomic experiments (e.g. ref. [7]), targeted drugs generally induce changes in their primary targeted pathway, as well as secondary pathways (Fig. 4, yellow circles). At this point, it is not known whether drugs exist that induce transcriptional changes entirely outside of the L1000

gene space or which would only be detected in other layers of data (e.g. protein profiles). We regard this as a possibility and expect that future databases that span the entire transcriptome or substantial parts of the proteome will have increased power to disease signatures in neuroblastoma and other cancers.

The zebrafish xenograft system based on GFP-tagged NB-PDX3 cells developed herein provides a complementary tool to assess candidate therapeutics in neuroblastoma. Implemented in an automated 96-well format, it provides toxicological information, as well as a first assessment of tumour growth and invasion, as determined by the relative increase of observed GFP signal, with an assay time of 5 days. The assay should be used with awareness of the possible impact of reduced temperature conditions, and the intrinsic limitations of measuring tagged tumour cells by imaging methods. Since zebrafish embryos are small and carry their food supply up to 5 dpf, these short term studies require little handling. The assay thereby presents a complementary tool to prioritize candidate treatments for further studies in genetically predisposed animal models of neuroblastoma such as the *TH-MYCN* mice[48], the *LIN28B* transgenic mice[53] or the *Dβh-EGFP-MYCN* zebrafish harbouring coexpression of activated *ALK* and *MYCN*[54].

Our computational method extends and complements current tools for pharmacological data mining. Our specific goal has been to build a pipeline that takes risk factors as input and provides targets to the user. We solve this problem by a 3-step algorithm: (i) estimation of mRNA signatures directly into the 978 gene space of L1000, (ii) match scoring of mRNA signatures and drug-specific L1000 expression profiles and (iii) deconvolution of targets using STITCH5 data. Some of the individual steps have counterparts in existing methods, others do not. Two interesting search engines have been proposed that carry out a task similar to the second step of our pipeline: L1000CDS2 and CREEDS[18,19]. Both web tools match pre-defined gene expression signatures to pharmaco-transcriptomic databases. L1000CDS2 makes use of the L1000 compendium and CREEDS uses expression profiles manually extracted and curated from public repositories. Acknowledging that an increasing number of studies are exploring possible relevance of LINCS/L1000 or related data to disease[7], our methods will thus complement existing tools and adds an evaluated algorithm that helps the user through the multi-step process of linking risk phenotypes to therapeutic targets. Users of TargetTranslator can also opt to prioritize target discovery for specific cell lines by weighing their contribution to the pipeline. An example of this is given in (c.f. Supplementary Fig. 8). Looking ahead, emerging data from, e.g. the St Jude Cloud or recent paediatric cancer genome atlases[5] will likely strengthen the analysis further.

In summary, TargetTranslator improves our understanding of neuroblastoma interventions and provides a practical tool for drug discovery. Users can upload their own data, which makes TargetTranslator easy to extend to other research areas where phenotype-drug-target relations are of interest, e.g. in other types of cancer or in the reprogramming of cell states (compare ADRN/MES in this paper).

## Methods

**Data sources**. We obtained 835 neuroblastoma patient molecular and clinical data from the R2, TARGET and SEQC data sets (download links in Supplementary Methods). All mRNA data were log-transformed (log Affymetrix gene signal for R2 and TARGET, log(FPKM+1) for SEQC) and row centred. Clinical data, genetic aberrations were obtained for each data set and gene signatures of differentiation used markers (listed in Supplementary Data 1). Level 3 LINCS/L1000 data were processed by the Remove Unwanted Variation (RUV) algorithm to remove batch effects and pooling of replicates to yield drug-specific log2 fold change profiles for each drug, relative to vehicle (DMSO) control (Supplementary Methods and [55]). We used RUV to pool information across doses and time points, to yield integrated

fold change profiles for each drug that were used for analysis (Supplementary Methods and benchmarking in Supplementary Fig. 1B). We obtained protein–protein and drug-protein network data from string-db.org and stitch.embl.de defining a combined score above 900 of 1000 as a link (Supplementary Methods). The processed data can be accessed at targettranslator.org/downloads.

**TargetTranslator method**. Data from profiled and clinically annotated tumour samples are arranged into two types of matrices. The first, which we term the disease data, contains uni- or multivariate data, available for all patients, which is relevant for risk or outcome, i.e. all items in Table 2. The second contains the corresponding RNA data for these tumour samples, restricted to the gene set in LINCS/L1000. We denote disease data matrices by $\mathbf{Y}(i)$ and RNA-matrices by $\mathbf{Z}(i)$, for cohorts $i = 1, 2, \ldots$. The disease data matrices thus have dimensionality $\dim(\mathbf{Y}(i)) = p_{\text{disease}} \times n_i$, where the same set of $p_{\text{disease}}$ risk-associated factors are considered across all cohorts. The RNA data are of dimension $\dim(\mathbf{Z}(i)) = p_{\text{LINCS}} \times n_i$. We assume that the risk data in $\mathbf{Y}(i)$ can be summarized with a low-rank feature such that

$$\mathbf{Y}(i) = \mathbf{H}(i)\mathbf{F}(i) + \epsilon_Y(i), \qquad (1)$$

where $\dim(\mathbf{H}(i)) = p_{\text{disease}} \times k, \dim(\mathbf{F}(i)) = k \times n$. $\mathbf{F}(i)$ represents a $k$-dimensional feature across patients in cohort $i$ that summarizes the patient variability across patients with respect to the $p_{\text{disease}}$ outcome associated factors. TargetTranslator extracts these features through a low-rank matrix decomposition (SVD). Next, we project data $\mathbf{Z}(i)$ onto the extracted features $\mathbf{F}(i)$;

$$\mathbf{Z}(i) = \mathbf{B}(i)\mathbf{F}(i) + \epsilon_{\mathbf{Z}}(i) \qquad (2)$$

where $\text{di}(\mathbf{B}(i)) = p_{\text{LINCS}} \times k, \dim(\mathbf{F}(i)) = k \times n_i$. The matrix $\mathbf{B}(i)$ are the $k$-dimensional signatures for the L1000 landmark genes. These are obtained through the regression

$$\widehat{\mathbf{B}}(i) = \mathbf{Z}(i)\mathbf{F}(i)^T\left(\mathbf{F}(i)\mathbf{F}(i)^T\right)^{-1}. \qquad (3)$$

TargetTranslator can also be run in a supervised setting; here, the user provides the latent variables $\mathbf{H}(i)$, typically as gene signatures (c.f. Supplementary Data 1), whereby $\mathbf{F}(i)$ is found by least squares. In another special case, the method can accommodate left-censored survival data. For this, the feature $\mathbf{F}$ is defined to be the log-proportional hazard of each patient, as obtained by a Cox regression model (Supplementary Methods).

Next, we are concerned with the matching between the signatures $\widehat{\mathbf{B}}(i), i = 1, 2, \ldots$ and the full LINCS/L1000 compendium, which we have organised into a 3-dimensional table, in which the $p_{\text{LINCS}}$ rows are genes, columns $C = 19763$ are drugs (or shRNAs) and $S = 14$ layers are cell lines (we use the $N = 14$ cell lines in LINCS/L1000 with at least 1000 unique drugs). We use $\mathbf{g}_{r,s}$ to denote the (z-transformed) gene expression vector for drug $r$ in cell line $s$ and the function $\sigma(\mathbf{g}, \mathbf{h}) = \sigma(\mathbf{g}^T\mathbf{h})$ to denote the similarity between any two profiles, where $\mathbf{g}^T\mathbf{h}$ is the scalar product between $\mathbf{g}$ and $\mathbf{h}$. The explicit form for $\sigma(\cdot, \cdot)$ is given in the Supplementary Methods. Finally, we compute the match score of a perturbation $r$, is defined as:

$$S(r) = \bar{\sigma}(r)\frac{1}{N}\sum_{s=1}^{N}\sigma(\mathbf{g}_{r,s}, \pm \mathbf{B}(1)) \times \sigma(\mathbf{g}_{r,s}, \pm \mathbf{B}(2)) \times \ldots, \qquad (4)$$

where $\pm \mathbf{B}$ denotes that the score is computed either with respect to $-1 \times \mathbf{B}$, to detect drugs that suppress the signature, or $+1 \times \mathbf{B}$, to detect drugs that enhance the signature. $\bar{\sigma}(r)$ denotes the drug-specific propensity for signatures to match across cell lines, estimated from LINCS/L1000 data (precise definition in the Supplement). This is weighed together with the average (expected) similarities between LINCS profiles and the TargetTranslator signatures $\mathbf{B}(i)$ across all cohorts. The resulting match score has three key properties. First, since each of the $\sigma$ terms has a value between 0 (no match) and 1 (match at a level expected for the same target), the match score will be on the interval between 0 and 1. Second, the $\bar{\sigma}$ term specifically serves the purpose of giving a stronger weight to any perturbation that gives a consistent response across model cell lines. Third, since a product is formed between the $\sigma$ values across all cohorts, this score will emphasize matches that are consistently observed across cohorts. Permutation simulations are described in the Supplementary Methods, benchmarking in (Supplementary Fig. 1B–D).

For target deconvolution, we consider the relationship between the aggregate scores $S(r), r = 1, 2, \ldots$ of the different perturbations; and the pathway context or target of each perturbation. For a given protein target, we use the data in STITCH to divide all perturbations into two sets: R+ and R−. The set R+ contains all perturbations with a STITCH score >900 (this is considered a stringent hit, whereas 700 is significant[8]). The set R- contains all perturbations below that threshold. We subsequently apply a two-sample, one-tailed, Kolmogorov–Smirnov test to test the sample $\{S(r), r \in R+\}$ vs the sample $\{S(r), r \in R-\}$. P-values are corrected by mafdr to obtain FDR q-values.

**Compounds and cell culture**. High scoring compounds chosen for further analysis were obtained from Selleckchem (omipalisib, torin-2, palbociclib, AZD5438, rosiglitazone, fasudil, JQ1, GW842166X, otenabant), Santa Cruz Biotechnology (DL-PDMP), Sigma-Aldrich (GW405833, PK11195, AS601245, retinoic acid,

SR144528), Tocris (ACEA, HU308, JWH133), MedChemExpress (CC-930, SP600125, D-JNKI-1, JNK-IN-8) and Enzo (lovastatin). All compounds were dissolved in DMSO or MilliQ water (palbociclib) according to the vendor's instruction to a stock of 10–20 mM. Patient-derived xenograft (PDX) cell lines NB-PDX2 and NB-PDX3 (termed LU-NB-2 and LU-NB-3 in previous publications) were cultured as described on LN-521 (Biolamina) coated primaria plates (VWR) in defined, bFGF/EGF supplemented medium[56,57]. NB-PDX2 derives from a cerebral metastasis of a stage 4 neuroblastoma patient and NB-PDX3 derives from a primary tumour in the adrenal gland of a stage 3 patient; both are MYCN amplified with 1p loss and 17q gain. SK-N-BE(2) (ATCC®CRL-2271™) and SK-N-SH (ATCC®HTB-11™) neuroblastoma lines were grown in the same medium as the PDX lines but on uncoated primaria plates. U3013MG glioblastoma cells from the HGCC biobank[58] were grown in bFGF/EGF supplemented medium[58,59]. NB-PDX2 and NB-PDX3 cells (Figs. 4–7) were authenticated by SNP profiling (Multiplexion, Germany). SK-N-BE(2) cell lines (Fig. 7) and U3013MG were confirmed by STR profiling. SK-N-SH (used in Fig. 5a) were not authenticated. Cells were tested for mycoplasma contamination on a regular basis. All cells were cultured under 5% $CO_2$ pressure at 37 °C to maximum confluency of 80% and detached using StemPro Accutase (ThermoFisher Scientific) during passaging.

**Pharmaco-transcriptomics of predicted drugs.** The full protocol is provided in the Supplementary Methods. In outline, cells were seeded 1 day prior to treatment in LN-521-coated 384-well microplates (BD Falcon Optilux #353962), at a density of 2000 cells per well. Drugs and vehicle controls were added by an automated workstation to implement a design that consisted of 16 treatments (13 compounds, 2 DMSO concentrations, 1 untreated cell medium) × 3 doses (equivalent to IC50, IC20 and IC10 (Supplementary Table 2)) × 4 replicates × 2 cell lines (total 384). The RNA-Seq libraries were prepared in a 384-well plate format using the SMART-Seq2 method[60] and sequenced on an Illumina HiSeq 3000 instrument for 51 cycles, targeting around 0.8 million reads per well as an output, which is sufficient to probe the transcriptome at the level of principal components, gene signatures and pathways[61,62]. High-quality reads were mapped against the Ensembl Homo sapiens (GRCh38, ftps://ensembl.org/pub/release-90/fasta/homo_sapiens/dna/). The uniquely-mapped reads aligned to exons were counted by HTSeq v.0.6.1[63], then normalized by the DESeq2 R package v.1.14.1[64].

The principal component analysis (PCA) plot in (Fig. 4) was computed from drug-induced log fold change values, pooling replicates for each dose. Receiver operating characteristic (ROC) curves based on comparison of (i) the correlation between neuroblastoma risk signatures and each drug's average fold change signature in LINCS/L1000 (here seen as the prediction) and (ii) the correlation between neuroblastoma risk signatures and each drug's corresponding average fold change signature in our RNA-Seq experiment in PDX cells (here seen as the ground truth). ROC curves (x = 1-specificity of prediction in relation to ground truth; y = sensitivity) and area under the curve (AUC) integrals of ROC curves were computed using Matlab's roc.m and trapz.m functions. Gene Set Enrichment Analysis (GSEA) was performed using the GSEAPreranked option in the GSEA software (version 2.0.9)[65]. For each compound, a ranked list was prepared by running differential expression analysis on transcriptomic data from treated and control samples using the DESeq2 R package v.1.14.1[64]. GeneIDs and log2 fold change values were extracted from the analysis output and used to create the ranked list. These were then submitted to GSEA in the pre-ranked mode using default parameters.

**In vitro evaluation of predicted drugs.** Cells were seeded on the day before treatment, followed by 72 h of drug exposure. Viability was detected using the metabolic activity assay AlamarBlue (Invitrogen), according to the manufacturer's protocol, and standardized by vehicle controls. Doses were chosen to include the TargetTranslator predicted dose. Dose–response sigmoidal curves were fitted using GraphPad Prism and used to select doses corresponding to the IC10, IC20, IC50 and IC80 concentrations for subsequent experiments. One-way ANOVA and Dunnett's multiple comparisons test were used to compare all neuroblastoma cell lines against the non-neuroblastoma line U3013MG.

For western blotting, 500,000 cells were seeded in LN-521 (Biolamina) coated 6-well Primaria plates (VWR), treated for 48 h with IC50 equivalent concentrations of positive controls (JQ1 and retinoic acid), as well as explored compounds. After lysis in RIPA buffer (ThermoScientific), 25 µg of protein was separated on a NuPAGE 4–12% bis-tris gel (Invitrogen) and transferred using the iBlot Gel transfer Stacks nitrocellulose mini kit (Invitrogen) on an iblot®gel transfer device (Invitrogen). Membranes were blocked with StartingBlocking T20 TBS (ThermoScientific) and stained with antibodies against the N-Myc (#ab16898, Abcam, 1:250) and with cyclophilin B (#ab16045, Abcam, 1:500) as a loading control. N-Myc protein expression was normalized against cyclophilin loading control and analysed as a ratio to non-treated control using a one-sample t-test with Benjamini–Hochberg false discovery rate (FDR) correction.

For apoptosis analysis, 10,000 NB-PDX2/3 cells were seeded on LN-521 coated 96-well primaria plates and treated with 0.1% IncuCyte Caspase-3/7 Green reagent for apoptosis (EssenBioscience #4440). Cells were treated with palbociclib 0.1 µM, lovastatin 3 µM, fasudil 50 µM, GW405833 6 µM, omipalisib 0.02 µM, AZD5438 7 µM, JQ1 3 µM, DL-PDMP 50 µM, retinoic acid 120 µM, PK11195 90 µM, torin-2 0.04 µM, DMSO 0.5%. Phase-contrast and fluorescent images were acquired over

96 h with the IncuCyte Zoom instrumentation (EssenBioscience). Time-lapse confluence and apoptosis were automatically masked using the built-in software and apoptotic response was quantified as apoptotic area (µm² of image) per confluence (% of image) and shown as a ratio to vehicle control.

To analyse confluence and differentiation, we collected and analysed phase-contrast images of NB-PDX2 or NB-PDX3 cells (2000 cells per well) plated in LN-521-coated 384-well microplates (BD Falcon Optilux #353962). Images were taken every 2 h for a total of 3 days, using a design with two replicate wells for each 2 cell lines, 7 doses and 11 compounds. The images were analysed by an in-house pipeline (Supplementary Methods, Supplementary Fig. 9) based on the U-net[66] convolutional neural network (CNN) for semantic segmentation. The CNN was trained to find cell protrusions as well as cell confluency. The number of protrusions of length > d in an image is roughly distributed as an exponential distribution. Accordingly, we fit the function $A \cdot e^{-k \cdot d}$, where $A$ is the number of protrusions, $d$ is the length and $k$ is the decay, for each image. We define:

$$Y_{np} = \frac{A}{N_{bp}} \tag{5}$$

$$Y_{md} = -1000 \cdot k \tag{6}$$

where $N_{bp}$ is the total length of the cell-background interface, $Y_{np}$ is the normalized amount of protrusions and $Y_{md}$ is the morphological differentiation score and describes the relationship between long and short protrusions in the image. Cells with long protrusions will hence have a larger morphological differentiation score. The confluence and protrusion data were quantified and assessed statistically using linear mixed effects models and bootstrapping, as described in the Supplementary Methods.

**Neuroblastoma zebrafish xenografts.** All zebrafish experiments have been approved by the regional animal ethics board (C68/15, 5.8.1-08213/2017, EP 161/14) and the conducted zebrafish experiments comply with all relevant ethical regulations for animal testing. Drug tolerance was measured by exposure of 2-day post fertilization (dpf) zebrafish embryos to water containing drugs with IC20, 50 and 80 levels of drug based on the viability data above. The level of toxicity was noted on a numerical scale, where 0 = no toxicity, 1 = toxicity noted only in one or two fish, 2 = weak signs of toxicity in several fish but no deaths, 3 = strong signs of toxicity but no deaths, 4–5 = lethal doses of drugs, with 5 suggesting that the effect might have been instant.

NB-PDX3 cells were transfected with a lentiviral construct containing GFP, luciferase and puromycin resistance genes (pBMN(CMV-copGFP-Luc2-Puro), Addgene plasmid # 80389, a kind gift from Prof. Magnus Essand, Uppsala University) and selected under puromycin treatment, following FACS sorting to obtain GFP expressing cells which were cultured as described above. Prior to embryo injection, cells were detached with Accutase, passed over a 40 µm sterile filter, pelleted, and resuspended in minimal media. Using a procedure described in detail in the Supplementary Methods, anesthetized 1 dpf fish were injected with 100-200 GFP-tagged neuroblastoma cells into the midbrain. Xenografted zebrafish were kept in E3 water at 33 °C. At 2 dpf, zebrafish were screened for tumour engraftment, manually sorted in 200 µl of E3 water into 96-well plates (1 per well), anesthetized, and imaged dorsally and laterally (×10 objective) for GFP baseline signal with a Vertebrate Automated Screening Technology (VAST) BioImager (Union Biometrica) using a LEICA DM6 B microscope and a Leica DFC9000 camera. Each drug was diluted in an additional 100 µl of E3 water and added to yield the desired end concentration. On day 5, zebrafish were sacrificed with 2.4 mM tricane and an endpoint image was acquired.

Zebrafish embryos were fixed in formaldehyde (PFA) and embedded in paraffin. Stainings were done on 6-µm sections, either with hematoxylin and eosin using standard procedures or with immunofluorescent co-staining of MKI67 (#M7240, DAKO), NuMA (#ab97585, Abcam) and DAPI (#00-4959-52, ThermoFisher Scientific). Stained zebrafish were imaged on an AxioImager (Zeiss). The detailed protocol is in Supplementary Methods.

**Neuroblastoma mouse xenografts.** All mouse experiments were approved by the regional ethics committee for animal research (N 231/14) and the conducted mouse experiments comply with all relevant ethical regulations for animal testing. For the xenograft study, SK-N-BE(2) cells ($15 \times 10^6$) were injected subcutaneously into the flanks of immunodeficient nude mice (female 5–6-weeks-old NMRI-nu/nu, Taconic). Mice with palpable tumours (0.20 cm³ or greater) were randomized to receive either GW405833 (45 mg per kg, n = 10), AS601245 (90 mg per kg, n = 8) or vehicle (n = 12) intraperitoneally for 8 days, or for 2–8 days. At the endpoint, tumours were dissected in smaller parts and either frozen or fixed in 4% PFA, for further analyses. Additional details and statistics are provided in Supplementary Methods.

Paraffin-embedded mouse samples from all treatment groups (n = 3–5) were cut into 6 µm sections and stained with hematoxylin and eosin (HE) or immunostained against proliferation marker MKI67 (#M7240, DAKO) and apoptosis marker cleaved PARP (#5625, Cell Signaling Technology) using a standard immunohistochemistry protocol. Antigens were retrieved in citric acid-based buffer (#H-3300, Vector Laboratories) containing 0.05% Tween-20 in 2100 Antigen Retriever (Aptum Biologics Ltd.), sections were blocked with Animal-free

blocking solution (#15019, Cell Signaling Technology), primary antibodies were diluted in normal antibody diluent (Immunologic), BrightVision Goat Anti-Rabbit or Anti-Mouse HRP Polymer (Immunologic) was applied and signal visualized using BrightDAB (Immunologic). Sections were counterstained with Mayer's hematoxylin (Histolab) and mounted with Pertex (Histolab). Slides were scanned using Nanozoomer S60 (Hamamatsu) and representative images were taken with ×40 magnification. The number of positive nuclei was assessed using ImmunoRatio[67] plugin for ImageJ (NIH). From each section 10–15 images were taken, covering at least 50% of the total section area. Results were presented as a percent of positive nuclei.

**Reporting summary**. Further information on research design is available in the Nature Research Reporting Summary linked to this article.

## Data availability
The sequencing data that support the findings of this study have been deposited in the Gene Expression Omnibus (GEO) with the accession code GSE120920. The source data underlying Figs. 4g, 5, 6b, e, 7 and Supplementary Figs. 4, 5, 6, 9, are provided in the Source Data file. Signatures used in the manuscript are defined in Supplementary Data 1. Additional data files, including the processed LINCS/L1000 and STITCH data can be accessed at targettranslator.org/downloads. For information on materials, contact SN (sven.nelander[at]igp.uu.se).

## Code availability
An R version of the Targettranslator server is available on GitHub https://github.com/Targettranslator/targettranslator. For question regarding code, contact (sven.nelander[at]igp.uu.se).

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

## Acknowledgements

We thank Dr. Mauro Dacasto and Dr. Mery Giantin for providing access to their computational cluster. We thank Prof Björn Nilsson for valuable comments on the manuscript. We thank Wasifa Kabir for help with cell experiments, Dr María Victoria Ruiz Peréz for valuable help with marker gene lists and Dr Marek Skupiński for assistance with illustrations. We thank the Swedish Childhood Cancer Foundation, Swedish Cancer Society, Swedish Research Council and the Swedish Strategic Research Foundation for financial support. Open access funding provided by Uppsala University.

## Author contributions

S.N. conceived the project. E.A. and S.N. designed the study with support from C.K. and R.E. S.N., C.W., A.S. and R.J. developed the computational method. E.A., N.H., R.E., designed and performed the cell experiments. R.E. and M.V. performed the RNA pro-filing. MAH developed marker gene lists for Diff2 and Diff3. E.A., N.H. and S.N. developed the zebrafish assay. E.R. performed the image analysis and statistical analyses. T.K.O., C.D., P.K., M.D. performed the mouse study. S.P. and D.B. developed the NB-PDX cell cultures. E.A., R.E. and S.N. drafted the figures. R.E. and I.L. performed the GSEA. E.A. and S.N. wrote the first version of the paper.

## Competing interests

The authors declare no competing interests.
