## [Peer Review File · Nature Communications]

Reviewers' comments:

Reviewer #1 (Remarks to the Author):

This manuscript presents a computational method to associate cancer risk factors (or clinical/molecular features) to potential therapeutic targets. It used neuroblastoma as an example and verified the hypothesis in preclinical models. Overall, it is well-written, and the computational method is novel. It would demonstrate the power of integrative analysis of big data in therapeutic discovery. A few comments are as follows.

- 1) It is known that drug-induced gene expression profiles are highly dependent on biological context and treatment conditions (i.e., dosage and treatment duration) (PMID:28699633). The model only considered the effect from cell line, yet ignored the effect from dosage and treatment duration. Two profiles with a very high concentration regardless of corresponding drugs could be very similar.
- 2) The detail of the LINCS dataset preparation was missing. The dataset might be processed in their previous work, but it would be very helpful to introduce the process briefly. The current LINCS dataset covers much more compounds and cell lines than the one used in this study
- 3) It would be nice to elaborate on the matching method biologically. It seems that the model computes the similarity between disease signatures and drug signatures, but this field often groups the effect into a reversal or a mimic. How is the matching method different from others?
- 4) Disease signatures and drug signatures were derived from different platforms (one using RNA-Seq and another using LINCS own profiling technology), how did the model take this variation into account while matching them? What level of LINCS data was used? Were the profiles normalized with control samples already? What's the value of disease samples (TPM, FPKM or others)?
- 5) Would L1000 be sufficient to exploit in this task? Would any targets be missing because of the limited coverage of the transcriptome?
- 6) While mapping compounds to targets using STITCH, how did the authors deal with the direction (agonist or antagonist effect)? For example, if the inhibitors and activators of ER distribute on both sides, the enrichment analysis could barely identify ER as an enriched target.
- 7) STITCH only covers a small portion of drug-target relationships, would this limitation affect the enrichment analysis?
- 8) CNR2, the target highlighted in this manuscript, was not actually discovered from the KS test because of the small drug size. Need to justify this.
- 9) In Fig 3D, several drugs modulate multiple pathways, suggesting these drugs may not be selective against the specific target. It is known that many chemical probes are dirty (PMID: 28810148). For example, AS601245 selected for validating MAPK8 in this study has more than five targets according to STITCH? How did the authors justify the selectivity of these compounds? Functional studies such as knock-down experiment would be expected to confirm therapeutic targets,
- 10) The toxicity of the four drugs on page 11 might be because their IC50 are very high (three of them have IC50 over 30um in table S3). With such high doses, it is making sense that they could affect normal cells as well.

Reviewer #2 (Remarks to the Author):

In this manuscript, the authors present a new algorithm (TargetTranslator) which allows integrative analysis to predict how targeted drugging will affect mRNA signatures associated with high risk of disease. In vitro and in vivo experiments point at 2 novel candidates for targeted treatment of high-risk neuroblastoma, i.e. CNR2 and MAPK8.

While some possibly interesting results are presented in this manuscript, I have some major concerns about the used approach. I don't follow the reasoning why mRNA signatures (developed in the context of different research questions in the context of neuroblastoma biology) are the best

starting point for the identification of new drug targets in this disease. This should be more convincingly explained. Directly working with the survival data of the patients could be more valuable? The authors wrongly annotate these signatures as 'risk factors' (in the text and in Figure 1C) while most of them have never been shown to be risk signatures (e.g. ALK mutation status is not a risk factor).

Which data from R2 were used? There is an RNA-sequencing dataset of almost 500 NB tumors available (GSE49711). Why was this valuable dataset not used?

Page 3 and Figure 1: More information is needed concerning the different signatures (what specific lists (+ how many genes) from these publications are used) and in Figure 1 the references to these publications should also be added. It should also be explained better what is meant by "hazard ratio/survival"?

This manuscript needs more convincing data that demonstrate that the algorithm used in TargetTranslator lead to better results compared to existing methods. Moreover, more validation/comparisons are needed than the correlation analysis (as visualized in Figure 2A) to show that this approach works better than other approaches, e.g. does this approach identifies less/more/better candidate targets than other methods, etc?

It is not clear to me what values from the R2 and TARGET database are correlated to obtain the correlation coefficient visualized in Figure 2C.

Figure 2B: only 10 reproducible signatures were included in the PCA analysis: how were they defined and selected? Why was it (Figure 2A) not reproducible for the other signatures and which other methods should be used (as mentioned in the discussion section).

Reference on page 6: Figure 2E should be Figure 2D.

Table1B includes also targets with q-value < 0.05, but with no evidence for effect in neuroblastoma according literature? This is not very clear. If I understand well, Table 1A contains the targets with evidence in neuroblastoma according to literature (but not clear from legend)? How was the evidence for effect in neuroblastoma defined?

Figure 3B: this is very difficult to interpret. Authors should use another visualization/analysis to convince that the transcriptional change magnitude was proportional to dose and time.

Reviewer #3 (Remarks to the Author):

The study by Almstedt et al., has generated a new computing algorithm, called Target Translator, to integrate data from patient information with drug databases and cellular signaling networks to identify putative targets for high risk neuroblastoma patients. Targets and drugs are tested using cell line and zebrafish xenograph approaches, scoring outcomes such as gene signatures for relative risk, differentiation status and tumor growth in vivo. The authors have identified promising new drug targets and compounds for the treatment of high-risk neuroblastoma, which is significant. The overall study appears to be well designed, however I have major concerns regarding the rigor and reproducibility of the xenograph assays in zebrafish which impact the conclusions:

1) The PDX-derived cells are grown in zebrafish at 33 degrees- this is not a temperature usually tolerated well by human cells and is expected to cause significant stress and/or apoptosis. The authors should demonstrate the cell lines are cold-tolerated in vitro before the transplantation experiment and show such cells display normal cell proliferation and/or cell death kinetics

compared to parental cells maintained at 37 degrees. Otherwise any differences in drug responses could be due to combined effect of the drugs with activated stress and/or apoptotic pathways.

2) The injection site is described as the lateral hindbrain. Neuroblastoma is derived from the neural crest-derived peripheral sympathetic nervous system, not the CNS, so it is not clear why this injection site was chosen? The appropriate orthotopic site for transplantation in the zebrafish is the interrenal gland (head of the kidney).

3) The zebrafish images in Figure 5 show tumor cells in the midbrain (optic tectum) and forebrain at 48 hpf, not the hindbrain- so it appears the injection technique is not precise enough to accurately measure tumor growth based on spread from the primary injection site. Also, the images in Figure 5D are taken from different oblique and dorsal perspectives- the images should all be the same orientation, preferably both a dorsal and lateral image from the same fish at the 2 time points.

4) The authors state that 100 cells were injected per embryo, but do not describe how this was measured. This is an important procedure to describe.

5) The extent of GFP fluorescent after 5 days is not an accurate way of measuring tumor growth, as fluorescence could be affected by site of injection (see above), number of cells initially injected (see above) and movement of injected cells through interstitial and ventricular spaces (clearly evident in Figure 5E). A second, more direct method for quantifying tumor growth should be employed.

6) The Ki67 staining is a major concern. The figures show Ki67-positive (red) cells that are not GFP positive, which suggests either the staining is non-specific (Ki67 has not been shown to label zebrafish cells) or that there are GFP-negative cells injected (most likely). It is therefore not possible to use the total area of GFP spreading as a surrogate for tumor growth as there may be non-GFP cells that are not being counted in this assay.

7) The Ki67 staining should be performed on section tissue to avoid off-target staining.

8) Overall, these assays are still cell line-based assays in an animal, whereas an in vivo assay would require treatment of primary tumors in zebrafish or mice models or primary human PDX's that have never been exposed to plastic. The authors should limit the text/description to more accurately state this is a cell line-based xenograph assays.

9) There are no experiments showing the drugs are actually working on their expected targets in the tumor cells after treatment in fish.

10) The zebrafish and mouse MYCN-driven models of neuroblastoma are readily available, so the impact of the study would be significantly increased if the authors used these primary tumor models to show an impact of their drugs on tumor growth.

RESPONSE TO POINTS RAISED BY REVIEWERS

Reviewer #1

This manuscript presents a computational method to associate cancer risk factors (or clinical/molecular features) to potential therapeutic targets. It used neuroblastoma as an example and verified the hypothesis in preclinical models. Overall, it is well-written, and the computational method is novel. It would demonstrate the power of integrative analysis of big data in therapeutic discovery.

Response. Thank you for the kind words! We are glad to hear that our results are of interest.

1) It is known that drug-induced gene expression profiles are highly dependent on biological context and treatment conditions (i.e., dosage and treatment duration) (PMID:28699633). The model only considered the effect from cell line, yet ignored the effect from dosage and treatment duration. Two profiles with a very high concentration regardless of corresponding drugs could be very similar.

DONE. Thank you for the comment. We agree that biological context, treatment duration, and compound dose are all important determinants of the observed RNA expression response in drug-treated cells. We have included additional analyses as follows:

First, as the reviewer suggested, we have now successfully extended the TargetTranslator algorithm to pool results across doses and time points. The most principled way to achieve this in our algorithmic framework is to pool observations at the level of data preparation, for which we use the RUV model (Gagnon-Bartsch and Speed, Biostatistics 2012). By using dose and time as covariates, we obtain a combined effect for the transcriptional effect of each drug (revised Methods, Supplementary Methods). We have also added benchmarking results to show that this method compares favorably both to (i) individual time points or doses, and (ii) connectivity scores aggregated over doses and times using the RGEN metric (Chen et al, Nat Comm 2017, cited) (revised Figure S1). We thank the reviewer for this valuable suggestion of an extension.

Second, our original submission contained a systematic statistical analysis of dose and time in our follow-up experiments, where we used linear models to show significant dose-dependent and time-dependent effects (revised Table S1). In this experiment, drug effects (observed as log fold-change patterns) were quite consistent (in terms of altered gene expression) for the same drug across doses and time-points, and we did not see any cases where high doses led to similar responses for different drugs (c.f. Figure 3).

Specific revisions:

- Extended computational method to incorporate data across doses and time points (page 13, Supplementary Methods pages 3-6)
- Revised Figure S1B-D to support that pooling increased performance and to relate our method to pooled connectivity scores.
- Revised discussion to emphasize further that in our experiments the direction of transcriptional response was consistent, but that its magnitude was dose- and time-dependent (page 12).

2) The detail of the LINCS dataset preparation was missing. The dataset might be processed in their previous work, but it would be very helpful to introduce the process briefly. The current LINCS dataset covers much more compounds and cell lines than the one used in this study

Thank you for the comment. We have now included more information about data preparation (above) and substantially extended the number of drugs analysed. While the original submission was based on a subset of the

data (7053 drugs), we now use the full available L1000 (GSE92742) and the number of pharmacological agents used is now extended to 19763. The InChi key annotation recently provided by the Broad Institute even further facilitates integration with datasets like STITCH. As the reviewer is aware, the LINCS/L1000 study has a rather unbalanced design (the number of perturbations/drugs per cell lines differs across the 77 cell lines). We have chosen to incorporate data from the 14 cell lines with data from at least 1000 unique drugs (Supplementary Methods, page 5), which gives a good trade-off- between the number of drugs and number of cell lines. We thank the reviewer for the valuable suggestion to extend the dataset used.

Specific revisions:

- Revised and extended description of LINCS/L1000 data preparation (Supplementary Methods, pages 4-5)
- Substantially extended analysis pipeline to involve 19763 drugs, compared to 7053 in the first submission, with consistent results for key hits (Figure 1, Table 1).

3) It would be nice to elaborate on the matching method biologically. It seems that the model computes the similarity between disease signatures and drug signatures, but this field often groups the effect into a reversal or a mimic. How is the matching method different from others?

DONE. Thank you for this comment. We have now added explanations in the Results and Methods sections to clarify that our analysis does take the positive/negative match into account, as defined by the sign operator in equation 4. Biologically, we are searching for drugs that are reversing/anti-correlating against signatures associated with disease risk/oncogenes (e.g. MYCN) and enhancing/correlating with signatures associated with differentiation (a therapeutic goal in NB treatment).

Specific revisions:

- This is now further clarified, in Results (Page 3 and Page 5) and Methods (page 15).

4) Disease signatures and drug signatures were derived from different platforms (one using RNA-Seq and another using LINCS own profiling technology),

- a) How did the model take this variation into account while matching them?*
- b) What level of LINCS data was used?*
- c) Were the profiles normalized with control samples already?*
- d) What's the value of disease samples (TPM, FPKM or others)?*

DONE. Thank you for these comments.

- a) Conceptually, the key principle is to first estimate log fold-change differential expression profiles from the L1000/LINCS data (drug-treated vs DMSO control). Similarly, the disease signatures are on a log scale, e.g. MYCN amplified vs non-amplified cases, which means that drug and disease signature are compared in a relevant RNA space in equations 1-4 (page 14-15). Empirically, the study contains checks of cross-platform consistency. In Figure 2A, we confirmed the consistency signatures derived from each cohort. In Figure 3, we confirm that drug-specific effects on signatures predicted from L1000 were observed by RNA sequencing in neuroblastoma cells. For further validation of L1000, we refer to (Subramanian et al, Cell, 2018).
- b) We use the level 3 LINCS data (quantile-normalised expression values).
- c) Based on the level 3 data, our RUV normalisation (above) estimates log fold-change in relation to untreated controls.
- d) The type of expression value is now clarified in Methods. The R2 data is Arraybased, and used in its provided normalised logged form. All RNAseq data (TARGET, GSE47911, our own experiments) are FPKM based, and logged. The datasets agree well as shown by our cross-validation of signatures in Figure 1.

Specific revisions

- We have now clarified these aspects in the Methods (p13) and Methods supplement (pages 4-5).

5) Would L1000 be sufficient to exploit in this task? Would any targets be missing because of the limited coverage of the transcriptome?

DONE. Thank you for the comment. We agree that the ability of mRNA space, and L1000 gene space in particular as a proxy for neuroblastoma disease signatures is not self-evident, which is why we evaluated this in Figure 2A. A strong feature of our method is the projection of signatures into L1000 gene space. As a specific example, the average correlation of signatures for MYCN amplification in L1000 gene space is $r=0.89$, even if MYCN itself is actually not part of the L1000 panel. A second example is the 42 gene signature by dePreter et al, which has a correlation of 0.93, even if only 7 genes in that panel are L1000 markers. Extending the original submission, the consistency analyses are now based on 498 additional patients, and a total of 835 patients. There are also signatures that are not well captured by RNA proxies, regardless of the estimation method or the gene space (Figure 2A), which we do not recommend for L1000 analysis. The extensive experimental validation (Figure 3-6) presented further strengthens our approach.

Specific revisions:

- Improved explanation of the rationale behind an RNA-based matching approach (Pages 3, 5)
- Extended discussion of factors that may affect signature consistency; also, elaborate on the role that the effect that restriction to L1000 space plays in the analysis, including the possibility that some drugs might cause changes entirely outside of L1000 gene space (page 12)

6) While mapping compounds to targets using STITCH, how did the authors deal with the direction (agonist or antagonist effect)? For example, if the inhibitors and activators of ER distribute on both sides, the enrichment analysis could barely identify ER as an enriched target.

DONE. Great point. This is actually not a problem, for the following reason. First, note that our match score is computed separately for positive and negative matches (Results page 5 and in eq 4, page 15). When scoring for negative matches, an antagonist will have a score close to 1, and an agonist will have a score close to 0. In both cases they actually contribute to a low p-value of the KS test, since both low (close to 0) and high (near 1) values will shift the cumulative distribution distance that underlies the KS test. Similarly, when scoring for positive matches, agonists will have match scores close to 1 and antagonists be close to 0.

Revisions: this has been clarified in the Supplementary Methods (page 3).

7) STITCH only covers a small portion of drug-target relationships, would this limitation affect the enrichment analysis?

DONE. Thank you for the interesting comment. The human part of STITCH contains 15,473,940 associations between proteins and chemicals. Mapped to the LINCS/L1000 data, the STITCH data contains 452,782 links between 2126 unique human proteins and chemical compounds at score cutoff (>900) used in this work. While STITCH is a leading effort to compile known links, we expect that there should exist unknown links, that remain to be discovered. While it is hard to speculate how much will be gained as future versions of STITCH become available, it is at least possible to confirm that our statistical enrichment test used is robust to missing drug-protein data. To illustrate this, we ran TargetTranslator with (i) the full STITCH data, and (ii) subsampled versions of the STITCH data, in which a specific random percentage of links had been removed:

	Enriched targets ($q < 0.001$)
Full STITCH	88
10% removed	87 JUN lost

25% removed	83	AURKA, CHRM4, FLT3, JUN, PGR lost
50% removed	58	30 targets lost

Thus, our test is robust to missing data in STITCH, up to 25% deletion. The two reported new targets, MAPK8 and CNR2 remained even after deleting 25% (MAPK8) and 50% (CNR2) of links. We still expect the method to give stronger results as more data become available.

Specific revisions:

- The above analysis is described under Benchmarking in (Methods Supplement, page 6).

8) CNR2, the target highlighted in this manuscript, was not actually discovered from the KS test because of the small drug size. Need to justify this.

DONE. This has changed due to (i) incorporation of 498 additional patients, and (ii) addition of more than 10,000 compounds per the suggestion to incorporate a larger portion of L1000 (above). With the new and extended data, which gives stronger signals overall, CNR2 has a q-value of 0.0004 and is among the top 70 targets.

9) In Fig 3D, several drugs modulate multiple pathways, suggesting these drugs may not be selective against the specific target. It is known that many chemical probes are dirty (PMID: 28810148). For example, AS601245 selected for validating MAPK8 in this study has more than five targets according to STITCH? How did the authors justify the selectivity of these compounds? Functional studies such as knock-down experiment would be expected to confirm therapeutic targets.

DONE. Thank you for the comment. We have now extended the study with multiple experiments to address this, presented as panels in the new Figure S4. The two key targets of interest are CNR2 and MAPK8. The molecular pharmacology of cannabinoid receptors (CNR2 and its isoform CNR1) has a strong toolkit, with both agonists and antagonists against both isoforms (c.f. Soethoudt et al. Nat Comm 2017). Using several antagonists and agonists (c.f. Table 2), we show that neuroblastoma cells are selectively sensitive to CNR2 agonists at low concentrations (Figure S4A-B), and that this effect is selectively rescued by CNR2 antagonists (Figure S4C-D). Together with the extended zebrafish experiments and newly added mouse study (Figure 6) these data leave little room, in our view, to neglect CNR2 agonists as a possible neuroblastoma therapy.

MAPK8 was approached in a similar fashion, using multiple inhibitors to its pharmacologically relevant isoforms MAPK9 and MAPK10. In this case, however, even the most recent inhibitors have some cross-reactivity, but the results are consistent with MAPK8 as the target, as the MAPK9/10 selective inhibitor was inactive (Figure S5E-F). We corroborated the results with a meta-analysis of RNAi data from the Broad institute's dependency map project, showing that neuroblastoma cell lines are sensitive to MAPK8, but not MAPK9/10, knockdown compared to non-neuroblastoma cell lines (Figure S4G).

In Figure 3D, we performed a GSEA analysis of drug-induced changes in patient-derived neuroblastoma cells. We have marked the intended targeted pathway by a yellow circle. The general finding was that the intended pathway was altered, as well as other pathways. For instance, the highly specific MTOR inhibitor torin-2 (with a target affinity IC50 of less than 5 nM) produces an effect on the MTOR pathway, as well as e.g. G2/M checkpoint genes (Figure 3D). We believe this pattern to be expected. Even for drugs with a high single-target specificity, we can observe a complex change in RNA pattern that involves several pathways (c.f. Subramanian et al, Cell 2018). Another interesting case-in-point is MEK inhibitors, which are agreed to be specific to MEK1/2 isoforms, but still produce changes in several downstream pathways (e.g. Gysin et al, Mol Cancer Res 2012).

Revisions:

- Added experimental and computational studies to support target selectivity of CNR2 and MAPK8
- Clarify that RNA profiles in response to drugs are often complex and involve both the intended and other pathways (Discussion, page 12).

Reviewer #2 (Remarks to the Author):

1) I don't follow the reasoning why mRNA signatures (developed in the context of different research questions in the context of neuroblastoma biology) are the best starting point for the identification of new drug targets in this disease. This should be more convincingly explained. Directly working with the survival data of the patients could be more valuable?

DONE. Thank you for the comment. We have now even further clarified the reasoning in the introduction and results. The use of RNA signature readout in drug studies is well explained in e.g. (Subramanian et al, Cell 2018). Our paper identifies high-risk neuroblastoma as a promising candidate for integrative data mining, for two main reasons. First, several biological and empirical arguments suggest that many aspects of neuroblastoma can be detected as RNA signatures, e.g. MYCN-driven transcription, patient risk, or cell differentiation (e.g. van Groningen, et al. Nat. Genet. 2017, Ribeiro et al Cell Reports 2016, De Preter et al, Clin Cancer Res 2010, White et al, Oncogene 2005). Second, from a computational perspective, there are new major data assets that strongly warrant data integration, e.g. (1) survival data, (2) patient genomics data, (3) RNA drug profiling data, and (4) target-protein information. We demonstrate that a new algorithm can integrate these factors to detect new molecules effective in patient-derived cell based models of NB. This is a new approach, and one that is likely to gain importance as more data becomes available. To the extent that future data sets contain protein profiles (instead of RNA) of both patient samples and drug treated cells, TargetTranslator can be applied in such a setting as well, or using a combination of protein and RNA. Regarding the question about survival data, we point out that our method does incorporate survival data, but in order to be relevant for target identification it includes other data types, like the tumor RNA profiles. As a case-in-point, our discovery of CNR2 as a target specifically depends on this integrative strategy.

Specific revisions:

- Introduction and Results section revised to further clarify these points further (page 2, 3 and 5).

The authors wrongly annotate these signatures as 'risk factors' (in the text and in Figure 1C) while most of them have never been shown to be risk signatures (e.g. ALK mutation status is not a risk factor).

DONE. Thank you for the comment. We agree that the terminology was not optimal, as the analysis considers both bona fide risk factors as well as other data of biological interest (e.g. ALK and differentiation signatures). We have carefully revised Figure 1C and the text, using suitable alternative terms (e.g. 'disease signatures' instead of 'risk signatures') and qualifiers (e.g. 'risk factors and key oncogenes'), depending on context.

Specific revisions:

- Terminology updated throughout manuscript.

2) Which data from R2 were used? There is an RNA-sequencing dataset of almost 500 NB tumors available (GSE49711). Why was this valuable dataset not used?

DONE. This is a great comment. The reason that GSE49711 (termed SEQC by the authors) was not included in the original analysis was its comparatively limited genetic annotation of each case. But we agree that it contains a high number of cases and is therefore of high potential value to strengthen parts of the analysis, meaning that the study now analyses a total of 835 cases. We have therefore included the SEQC cohort in the revised version of the paper

and the derived signatures agree well with previous results obtained for the R2 and TARGET data (Figure 2A) and accordingly report results obtained for the full data (Table 1).

Specific revisions:

- All signatures in the paper are now computed across 3 instead of 2 cohorts, and the total number of cases is 835 instead of previous 337.

3) Page 3 and Figure 1: More information is needed concerning the different signatures (what specific lists (+ how many genes) from these publications are used) and in Figure 1 the references to these publications should also be added. It should also be explained better what is meant by "hazard ratio/survival"?

DONE. Thank you for pointing this out. We apologize that this information was not included, and have added it to Figure 1C. The revised Supplementary methods contains a detailed explanation of how the signatures were constructed, and a new supplementary table lists all genes that were used to define the signatures.

Specific revisions:

- Figure 1 and legend revised to include the above details.
- All gene signatures are added to Table S3.
- Added section 'preparation of neuroblastoma datasets and signatures' to Supplementary methods (pages 3-4)

4) This manuscript needs more convincing data that demonstrate that the algorithm used in TargetTranslator lead to better results compared to existing methods. Moreover, more validation/comparisons are needed then the correlation analysis (as visualized in Figure 2A) to show that this approach works better than other approaches, e.g. does this approach identifies less/more/better candidate targets then other methods, etc?

DONE. Thank you for the comments. As discussed above, we have now extended and improved these analyses with a systematic comparison to demonstrate the benefit of pooling data across replicates, doses, and time points and compared it to a state-of-the-art matching method, RGENES (See above). The benchmarking (Figure S1A-D) specifically addresses the challenge of extracting compounds with a similar target, which is a specific goal of our analysis. In addition, we have extended the comparison to the state-of-the-art characteristic direction method (Figure 2A) by adding the GSE49711 data (above). We point out that TargetTranslator is an *integrative* pipeline that links three aspects (signature construction, compound searching, and target deconvolution) which is more general than existing methods, and that our study conclusively demonstrates the feasibility of this novel approach in a concrete disease setting.

Specific revisions

- Added more benchmarking comparisons (Figure S1, Supplementary Methods pages 5-6).
- Extended data analysis (one more cohort) and explanations linked to Figure 2A.
- Clarify that the proposed method has a more general scope (Discussion p 13)

5) It is not clear to me what values from the R2 and TARGET database are correlated to obtain the correlation coefficient visualized in Figure 2C.

DONE. Thank you for the comment. It is our impression that this question is potentially about Figure 2A, which is discussed below, here he answer regarding 2C. The goal of TargetTranslator is to associate risk signatures to drugs and targets. As part of that analysis, it first scores each compound in L1000 to each risk signature. These scores, defined by Equation 4, page 15) take on values between 0 and 1, which are conveniently displayed as a cumulative distribution as in Figure 2C, solid line. The dashed line shows the null distribution, obtained by permutation control simulations. The conceptually important part of the analysis is that when compounds are subselected with respect to

a particular target, e.g. AKT1 (blue curve), the curve is shifted to the right, indicating an increase in the scores selectively for the AKT1 targets, and so on. This effect, quantified by a Kolmogorov-Smirnoff test, is what TargetTranslator uses to predict targets.

Specific revisions

- Updated text on page 4 and legend of Figure 2 to clarify further the above points.

6) Figure 2B: only 10 reproducible signatures were included in the PCA analysis: how were they defined and selected? Why was it (Figure 2A) not reproducible for the other signatures and which other methods should be used (as mentioned in the discussion section).

DONE. Thank you for the comment. We have now added a detailed explanation how signatures were defined and selected. In brief, signatures were defined by applying equations 1-3 (Methods, page 14) to three independent neuroblastoma cohorts (TARGET, R2, SEQC). For each of the items listed in Figure 1C (MYCN amplification and so on), we derive a signature in each one of the cohorts. As an empirical check, we then correlated the signatures to see that they are consistent between cohorts (Figure 2A). The reproducibility of a signature will depend on several factors, particularly (i) the choice of algorithm, (ii) the choice of markers, (iii) the amount of data and (iv) the underlying biology. Figure 2A shows that TargetTranslator has higher signature reproducibility than an alternative state-of-the-art algorithm, and that signature reproducibility is high also in L1000 gene space (which helps downstream analyses). Signature consistency are covered in the Discussion (page 11). Altogether, we believe that our consistency check of neuroblastoma signatures is a valuable analysis and contribution, which helps define which particular aspects of neuroblastoma are productively analysed at the level of RNA signatures. We also point out that our data are stronger than the initial submission, since one more cohort has been added.

Specific revisions

- Signatures used are listed in Figure 1C, and Table S3
- Extended discussion of factors that can determine signature consistency, page 11.

7) Reference on page 6: Figure 2E should be Figure 2D.

DONE. Thank you for noticing this error. The mistake has been corrected.

8) Table 1B includes also targets with q-value < 0.05, but with no evidence for effect in neuroblastoma according literature? This is not very clear. If I understand well, Table 1A contains the targets with evidence in neuroblastoma according to literature (but not clear from legend)? How was the evidence for effect in neuroblastoma defined?

RESPONSE. Thank you for the comment. This appears to be a misunderstanding, since Table 1 summarizes the predictions made by TargetTranslator, of targets with an association to at least one of the neuroblastoma disease signatures. We have improved the presentation to clarify this point even further. The table does not explicitly mark known targets, but some known targets and their current support are mentioned in the text.

Specific revisions:

- Updated legend of Table 1.

9) Figure 3B: this is very difficult to interpret. Authors should use another visualization/analysis to convince that the transcriptional change magnitude was proportional to dose and time.

DONE. Thank you for the comment. In the original submission, we analysed dose and time dependency using a linear model, which was presented in Table S2, which summarises results for the first 4 principal components. We

have now extended table S2 with plots from the linear model analysis, showing the significant trends for two of the columns. We think that the PCA visualization in 3B helps the reader gain an intuition for how drugs affect the neuroblastoma cells.

Specific revisions:

- Table S2 (dose and time dependency p-values) extended with plot examples of dose and time dependency.
- Improvements to explanatory text, emphasizing that the PCA in Figure 3B is for visualization, whereas the formal test is in Table S2.

Reviewer #3 (Remarks to the Author):

The study by Almstedt et al., has generated a new computing algorithm, called Target Translator, to integrate data from patient information with drug databases and cellular signaling networks to identify putative targets for high risk neuroblastoma patients. Targets and drugs are tested using cell line and zebrafish xenograph approaches, scoring outcomes such as gene signatures for relative risk, differentiation status and tumor growth in vivo. The authors have identified promising new drug targets and compounds for the treatment of high-risk neuroblastoma, which is significant.

RESPONSE. Thank you for the kind words! We are glad to hear that our work was well received.

The overall study appears to be well designed, however I have major concerns regarding the rigor and reproducibility of the xenograph assays in zebrafish which impact the conclusions:

1) The PDX-derived cells are grown in zebrafish at 33 degrees- this is not a temperature usually tolerated well by human cells and is expected to cause significant stress and/or apoptosis. The authors should demonstrate the cell lines are cold-tolerated in vitro before the transplantation experiment and show such cells display normal cell proliferation and/or cell death kinetics compared to parental cells maintained at 37 degrees. Otherwise any differences in drug responses could be due to combined effect of the drugs with activated stress and/or apoptotic pathways.

DONE. Thank you for the interesting comment. We have now extended the study with experiments that investigate the effect of temperature (37 vs 33 degrees) in our patient-derived neuroblastoma cells (Figure S5). As expected, NB-PDX2 and NB-PDX3 cells show a reduced Alamar Blue assay signal, consistent with a reduction of cell metabolism at 33 degrees (Figure S5A), but also a *reduced* apoptotic rate (Figure S5B). Over a 90 hours time interval, NB-PDX3 cells (which were employed in the zebrafish assay) grow to the same confluence at 33 and 37 degrees (Figure S5C). We also see that cells are still proliferating after 4 days after engraftment (Figure S5D). The response of NB-PDX2 and NB-PDX3 cells to GW405833 or AS601245 was not potentiated by 33 degree conditions (Figure S5E). In fact, cells were even partially protected against AS601245 at 33 degrees. These results, together with the mouse *in vivo* study (at 37 degrees) preclude against hypothermia as a central factor behind our observations.

Specific revisions:

- Figure S5 added, comparing cell growth and drug response under 33 and 37 degree conditions.
- Figure 6 added, showing significant treatment effects in mice (i.e. at 37 degrees)

2) The injection site is described as the lateral hindbrain. Neuroblastoma is derived from the neural crest-derived peripheral sympathetic nervous system, not the CNS, so it is not clear why this injection site was chosen? The appropriate orthotopic site for transplantation in the zebrafish is the interrenal gland (head of the kidney).

RESPONSE. Thank you for the comment. We agree that in terms of biological homology, injections into the interrenal gland would be desirable. There were two main reasons for this choice of the midbrain as the injection site. First, it enabled robust injection into large numbers of individuals. Because the interrenal gland is <0.1 mm³, injection to this site was not practical. We also considered yolk sac injections, as in the previous literature (PMID: 29515255), but found that our PDX-derived cell lines did not grow well in this site. Second, although neuroblastoma derives from the neural crest/peripheral sympathetic nervous system, 8 % of stage 4 neuroblastoma patients develop cerebral metastases after 3 year (PMID: 12833468). This indicates that high-risk neuroblastoma can grow in the CNS niche, despite the PNS origin.

Specific revisions:

- Explain more clearly the choice of injection site (page 9)

3) The zebrafish images in Figure 5 show tumor cells in the midbrain (optic tectum) and forebrain at 48 hpf, not the hindbrain- so it appears the injection technique is not precise enough to accurately measure tumor growth based on spread for the primary injection site. Also, the images in Figure 5D are taken from different oblique and dorsal perspectives- the images should all be the same orientation, preferably both a dorsal and lateral image from the same fish at the 2 time points.

DONE. Thank you for the comment. The reviewer is right that variations in observed GFP signal can arise from treatment but also from the site of injection, number of cells, and rotation/angle. It is therefore important to control for these factors. First, regarding the site of injection, we clarify that cells were injected into the midbrain of embryos, not the hindbrain as stated. We apologize for the error of nomenclature and thank the reviewer for pointing it out. To verify that the tumor location is consistent, we carried out an analysis of all injected fish (Figure 5B), showing that the tumor is localised to the midbrain in 60% of all embryos individuals, or both the mid- and hindbrain in 17% of fish, as well as other brain regions. Second, each fish is imaged at lateral and dorsal angles at two separate time points, both 2 days post fertilization (2 dpf) and at 5 dpf, so we can calculate the ratio of tumor involvement by forming the ratio of the two, and apply statistics in a sufficient sample of fish (Figure 5D-E). This way, we correct for variation in the number of injected tumor cells, as well as tumor location, controlling for both these factors. In addition to the improved zebrafish assay (Figure 5, S5), we have improved the presentation to make these points clear.

Specific revisions:

- Improved zebrafish assay based on VAST imaging and two separate time points (Figure 5,S5)
- Further discuss the properties of the zebrafish assay (page 12).

4) The authors state that 100 cells were injected per embryo, but do not describe how this was measured. This is an important procedure to describe.

DONE. Thank you for pointing this out. We have now improved the supplementary methods description to clarify the injection method, which is based on (Pudelko et al., Neuro-Oncology 2018). Clarifying further with this team, cells were counted manually three times by ejection of a fixed volume from the micro-injector into the plate to set the volume (less than 10 picoliter) to a corresponding 150 cells.

Specific revisions:

- Extended methods supplement to cover these aspects (Methods Supplement, page 10).

5) The extent of GFP fluorescent after 5 days is not an accurate way of measuring tumor growth, as fluorescence could be affected by site of injection (see above), number of cells initially injected (see above) and movement of injected cells through interstitial and ventricular spaces (clearly evident in Figure 5E). A second, more direct method for quantifying tumor growth should be employed.

DONE. Thank you for the comment. These sources of variation have been analysed and controlled for, as described above. Imaging-based monitoring of growing tumors, whether in experimental animals or in patients, should be done with an awareness of the fact that observable tumor extent is the result of both growth and invasion processes. This is now pointed out in the discussion. We have now extended the study with a mouse experiment (Figure 6), in which we use longitudinally observed tumor size to measure growth.

Specific revisions:

- Added chart to Figure 5 to clarify that fish are imaged at two time points, thus enabling the relative increase of observed GFP signal.
- Zebrafish assay discussed further on page 12, addressing the above property of image-based observation of growing tumors.
- A second mouse experiment has been employed (Figure 6).

6) The figures show Ki67-positive (red) cells that are not GFP positive, which suggests either the staining is non-specific (Ki67 has not been shown to label zebrafish cells) or that there are GFP-negative cells injected (most likely). It is therefore not possible to use the total area of GFP spreading as a surrogate for tumor growth as there may be non-GFP cells that are not being counted in this assay.

DONE. Thank you for the comment. In the original submission, we used LightSheet tomographic microscopy to detect Ki67, and agree with the reviewer that the results were noisy and contained background signal due to the technology used. We also clarify that the example figure in the original figure had brightness of the green channel (GFP) set in such a way that red cells came across negative in the green channel. We apologize for this unclear presentation. In the revised paper, we instead use standard and well-tested immunohistochemistry protocols to evaluate xenografted mouse tumors (Figure 6, S6). We also include a section of a zebrafish brain stained with Ki67 to demonstrate that NB cells in the fish proliferate at 33 degrees (Figure S5).

7) The Ki67 staining should be performed on section tissue to avoid off-target staining.

DONE. We agree. The revised paper contains IHC data for mouse injected tumors, covering both Ki67, cleaved-PARP.

8) Overall, these assays are still cell line-based assays in an animal, whereas an in vivo assay would require treatment of primary tumors in zebrafish or mice models or primary human PDX's that have never been exposed to plastic. The authors should limit the text/description to more accurately state this is a cell line-based xenograph assays.

DONE. Thank you for pointing this out. We have now updated the text and description of the cell lines to more accurately convey this distinction.

9) There are no experiments showing the drugs are actually working on their expected targets in the tumor cells after treatment in fish.

DONE. Thank you for the comment. As discussed above, the revised manuscript now contains experiments to show that the viability response of neuroblastoma cells is specific to CNR2 activation, as it is induced by multiple CNR2 agonists, blocked by CNR2 antagonists, and not achieved by corresponding CNR1 selective compounds (Figure S4). We have also added data with available state-of-the-art inhibitors and RNAi meta-analyses to analyze the effect on MAPK8 (Figure S4). Furthermore, the revised manuscript contains a second study in mice (Figure 6, Figure S6), with a significant reduction of tumor growth and significant induction of apoptosis determined by cleaved-PARP upregulation, which is fully consistent with the *in vitro* findings. These results bode well for extended work on CNR2 targeted therapy in neuroblastoma, which is reserved for future publications.

10) *The zebrafish and mouse MYCN-driven models of neuroblastoma are readily available, so the impact of the study would be significantly increased if the authors used these primary tumor models to show an impact of their drugs on tumor growth.*

DONE. Thank you for this comment. We have now conducted an extended *in vivo* treatment assessment in mice, conducted by Prof. Kogner's team at Karolinska Institutet. The longitudinal 8-day experiment showed that GW405833 administered i.p. at a dose of 45 mg/kg/day significantly decrease tumor growth in neuroblastoma (Figure 6A-6C), that postmortem tumor weight was reduced (Figure 6D) and that treated tumors had elevated fraction of apoptotic cells as determined by cleaved-PARP (Figure 6F-G). No toxic effects were observed upon GW405833 administration. Treatment of mice by AS601245 also had an effect on tumor growth, but showed signs of toxicity (Figure S6). As far as we know, this is the first report of *in vivo* effect of CNR2 agonists against neuroblastoma, and motivate further investigation, reserved for future work.

REVIEWERS' COMMENTS:

Reviewer #1 (Remarks to the Author):

The revision has addressed the majority of my comments. Given its overall high-quality, I am inclined to make the recommendation to publish if the following comments could be addressed.

- 1) The work reprocessed the dataset LINCS, which is publicly available. The authors should also release their processed dataset to the public along with the full list of predictions for future benchmarking.
- 2) They showed the substantial improvement of the pooled profiles, but did not discuss this critical point in the main text.
- 3) They still did not mention the number of compounds mapped to STITCH.
- 4) The name of Bayesian match score is confusing. Why is it called Bayesian without showing Bayesian statistics?
- 5) Fig 3c: I don't think the comparison is reasonable. Other than induced/suppressed effect, the majority should be no effect. Differentiating two extreme groups could be much easier than distinguishing effect vs. no-effect. Excluding the no-effect group could mislead the performance.
- 6) CNR2 could be a very novel target in neuroblastoma. It would be nice to show the connections between risk factors, signatures, drug hits and CNR2.
- 7) Figure 2A; Y axis label is incorrect

Reviewer #2 (Remarks to the Author):

The authors have answered on all my questions and adapted the manuscript accordingly.

Reviewer #3 (Remarks to the Author):

The authors have done an excellent job addressing my comments and have significantly increased the rigor of the in vivo assays and added mice experiments, which 1) allow more precise conclusions to be made and 2) increase the impact of the study. The improved image quality of the zebrafish experiments, description of methods and additional quantification represents the highest standard for drug discovery using zebrafish.

Rod Stewart

Response to reviewer comments

Reviewer #1 (Remarks to the Author):

The revision has addressed the majority of my comments. Given its overall high-quality, I am inclined to make the recommendation to publish if the following comments could be addressed:

- 1) The work reprocessed the dataset LINCS, which is publicly available. The authors should also release their processed dataset to the public along with the full list of predictions for future benchmarking.*
- 2) They showed the substantial improvement of the pooled profiles, but did not discuss this critical point in the main text.*
- 3) They still did not mention the number of compounds mapped to STITCH.*
- 4) The name of Bayesian match score is confusing. Why is it called Bayesian without showing Bayesian statistics?*
- 5) Fig 3c: I don't think the comparison is reasonable. Other than induced/suppressed effect, the majority should be no effect. Differentiating two extreme groups could be much easier than distinguishing effect vs. no-effect. Excluding the no-effect group could mislead the performance.*
- 6) CNR2 could be a very novel target in neuroblastoma. It would be nice to show the connections between risk factors, signatures, drug hits and CNR2.*
- 7) Figure 2A; Y axis label is incorrect*

DONE. We are glad to hear that our manuscript is of interest and thank the reviewer for the suggested additional improvements, addressed as follows:

- 1) We have made our processed version of LINCS dataset available, along with the list of predictions. The data can be downloaded from targettranslator.org/downloads.
- 2) This is now mentioned in Results on page 4, which says: *'To gain predictive power, we used a normalised version of the L1000 data, in which the transcriptional effect of a drug is estimated from multiple replicates (Supplementary Figure 1).'*
- 3) This clarification is now mentioned in Results on page 4, stating that STITCH mapped to LINCS/L1000 comprises *'452,782 links between drugs and protein targets, involving 3421 unique LINCS/L1000 drugs and 17086 unique targets.'*
- 4) We agree that our method is not fully Bayesian but rather inspired by this philosophy. Our score is approximately equivalent to the expected posterior probability of a match in all cohorts, across cell-lines. However, since we did not explicitly derive the posterior, we agree that name Bayesian score is confusing. We have changed the text accordingly, using the term "match score" throughout and have also updated the methods and Supplemental to clarify how the score is computed and what the rationale behind it is.
- 5) This appears to be a misunderstanding. Figure 3C does already show induced / suppressed effect vs no effect as suggested. The blue curve shows suppressed effect vs no effect, and the red curve shows induced effect vs no effect. (In other words, none of the curves shows induced vs suppressed, which we agree would not be less informative). We have clarified the text on page 7 to make this fully clear, saying *'for both positive associations (vs no association) and negative associations (vs no association).'*
- 6) We have added an illustration of the high ranking hits linked to CNR2, with association to each risk signature, added as Supplementary Figure 7.
- 7) Thank you for pointing this out, the Y axis label in Figure 2A has now been corrected.

Reviewer #2 (Remarks to the Author):

The authors have answered on all my questions and adapted the manuscript accordingly.

Thank you for reviewing our manuscript and the many constructive remarks.

Reviewer #3 (Remarks to the Author):

The authors have done an excellent job addressing my comments and have significantly increased the rigor of the in vivo assays and added mice experiments, which 1) allow more precise conclusions to be made and 2) increase the impact of the study. The improved image quality of the zebrafish experiments, description of methods and additional quantification represents the highest standard for drug discovery using zebrafish.

Rod Stewart

Thank you for reviewing our manuscript and the many constructive remarks.